# Local prediction-learning in high-dimensional spaces enables neural networks to plan

Christoph Stöckl [1], Yukun Yang [1] & Wolfgang Maass [1] ✉

Planning and problem solving are cornerstones of higher brain function. But we do not know how the brain does that. We show that learning of a suitable cognitive map of the problem space suffices. Furthermore, this can be reduced to learning to predict the next observation through local synaptic plasticity. Importantly, the resulting cognitive map encodes relations between actions and observations, and its emergent high-dimensional geometry provides a sense of direction for reaching distant goals. This quasi-Euclidean sense of direction provides a simple heuristic for online planning that works almost as well as the best offline planning algorithms from AI. If the problem space is a physical space, this method automatically extracts structural regularities from the sequence of observations that it receives so that it can generalize to unseen parts. This speeds up learning of navigation in 2D mazes and the locomotion with complex actuator systems, such as legged bodies. The cognitive map learner that we propose does not require a teacher, similar to self-attention networks (Transformers). But in contrast to Transformers, it does not require backpropagation of errors or very large datasets for learning. Hence it provides a blue-print for future energy-efficient neuromorphic hardware that acquires advanced cognitive capabilities through autonomous on-chip learning.

Planning and problem solving are fundamental higher cognitive functions of the brain[1]. But according to the recent review[2] it remains largely unknown how the brain achieves that. Planning is defined in this review as *the process of selecting an action or sequence of actions in terms of the desirability of their outcomes*. Importantly, this desirability may depend on distant outcomes in the future; hence some form of look-ahead is required for efficient planning. From the mathematical perspective many planning tasks and problem-solving tasks can be formulated as the task to find in some graph a shortest path from a given start to a given goal node, see the first chapters on planning and problem solving in the standard AI textbook[3]. This graph is in general not planar, and its nodes do not have to represent locations in some physical environment. Rather, its nodes represent in general a combination of external and/or internal states of a planning process.

Similarly, its edges do, in general, not represent movements in a physical environment. They can represent diverse actions such as acquiring information by reading the fine print of a document, getting a screw driver for the screw in front of the agent, or buying the offered product. In contrast to navigation tasks, the edges from a node may represent in the general case of planning and problem-solving actions that are specific to the current node, and may make no sense at other nodes.

According to ref. 2 we are lacking an understanding of how the brain plans and solves problems not only on the biological implementation level, but also on the two top levels of the Marr hierarchy[4]: the conceptual level and the level of algorithms. Since these higher levels need to be clarified first, we will focus on them. There are several ways of implementing these higher levels in neural circuits; one

[1]Institute of Theoretical Computer Science, Graz University of Technology, 8010 Graz, Austria. ✉e-mail: maass@igi.tugraz.at

example implementation is provided in the Supplementary Information. On the conceptual level we show that it suffices to learn a suitable cognitive map of the problem space. The concept of a cognitive map originated from experimental data on knowledge structures in the rodent brain. Earlier work focused on the representation of landmarks and pathways of a 2D maze in cognitive maps[5]. More recent experimental data suggest that these cognitive maps encode in addition information about relations between actions and locations[6,7]. Experimental data on the human brain show that it uses cognitive maps also for mental navigation in abstract concept spaces[8], see[8,9] for recent reviews. That cognitive maps in the brain encode relations between actions and the changes in sensory inputs which they cause had already been postulated before that[1,10,11]. But in spite of the rich inspiration from experimental data about cognitive maps, one was still lacking insight into neural network architectures and learning methods that are able to create cognitive maps that enable planning and problem solving.

With regard to this second level of the Marr hierarchy, the algorithmic level, we show that a simple neural network architecture and learning method suffices, to which we will refer as a Cognitive Map Learner (CML). The CML creates through local synaptic plasticity an internal model of the problem space whose high-dimensional geometry reduces planning to a simple heuristic search: Choose as next action one that points into the direction of the given goal. This is an online planning method according to the definitions in[2] and[3], since it is able to produce the next action with low latency, without first having to compute a complete path to the goal. Surprisingly, functionally powerful online planning algorithms have been missing even on the abstract level of AI, see[3]. It should be noted that the CML approach is not related to existing approaches to predict or replay sequences. The CML stores learnt knowledge in the form of a cognitive map, not in the form of sequences. The CML learns this cognitive map through self-supervised learning: by learning to predict the next sensory input (observation) after carrying out an action. Furthermore, it is able to recombine learnt experiences from different exploration paths, and can merge these experiences for physical environments with inferred knowledge or generalization based on inherent symmetries of physical environments. Altogether, the CML provides a new approach toward planning and problem solving on the conceptual and algorithmic level, not only for neuromorphic implementations.

CMLs share with Transformers[12] that they do not require a teacher for learning, and that the outcome of learning is an embedding of external tokens into a high-dimensional internal model. But in contrast to Transformers, the CML neither requires deep learning nor large amounts of data for learning: Its learning can be implemented through local synaptic plasticity, and it only needs a moderate amount of exploration. CMLs share their use of high-dimensional internal representation spaces not only with Transformers, but also with vector symbolic architectures (VSAs)[13–15]. VSAs have attracted interest both from the perspective of modeling brain computations, but also in neuromorphic engineering. However, CMLs are learning these high-dimensional representations, whereas previous work on VSAs relied largely on clever constructions of them.

Since powerful online planning methods are rare in AI, we compare the planning performance of the CML with the next more powerful class of planning methods in AI: Offline planning methods such as the Dijkstra's algorithm (see supplements Alg. 1) and the A* algorithm, see[3] for an overview. Surprisingly, we found that CMLs achieved for planning in abstract graphs almost the same performance as these offline planning methods. This is quite interesting from the functional perspective, since online planning with a CML is able to decide with much lower latency on the next step. In addition, CMLs can instantly adjust to changes of the goal, or to contingencies that make some actions currently unavailable. In other words, they capture some of the amazing flexibility of our brains to adjust plans on the fly. In contrast,

the Dijkstra algorithm and A* need to restart the computation from scratch when the start or the goal changes. Also common reinforcement learning (RL) methods are based on a given reward policy, i.e., on a goal that has been defined a-priori. Consequently, they need to carry out complex computations, such as value- or policy iteration, when the reward policy or the model changes.

We also consider cases where the problem space is not a general graph, but a 2D maze or a simulated legged robot. We show that the CML automatically exploits these structural regularities in order to generalize to unseen parts of the problem space. Since efficient approaches for learning to plan have not only been missing in theoretical neuroscience but also in neuromorphic engineering, we will briefly discuss at the end options for efficient implementations of CMLs in neuromorphic hardware.

## Results
### Fundamental principles of the cognitive map learner (CML) and its use for planning

We encode observations as vectors $\mathbf{o}$ in the $n_o$-dimensional space, and actions $\mathbf{a}$ as vectors in the $n_a$-dimensional space. By default, both are binary vectors with exactly one bit of value 1 (one-hot encoding); $n_o$ and $n_a$ represent the total number of possible observations and actions, respectively.

Observations from the environment and codes for actions are embedded by linear functions $\mathbf{Q}$ and $\mathbf{V}$ into a high-D continuous space S, see Fig. 1a. We will simply refer to S as the state space, though it also embeds actions. This is essential, because the CML builds through self-supervised learning in S a cognitive map that encodes relations between observations and actions. Note that the power of these linear embeddings can be substantially enhanced by combining them with a fixed nonlinear preprocessor that assumes the role of a kernel, a liquid[16], or a reservoir[17].

The goal of self-supervised learning by the CML is:

**Principle 1.** (Predictive coding goal)

$$\mathbf{Q}\mathbf{o}_{t+1} \approx \mathbf{Q}\mathbf{o}_t + \mathbf{V}\mathbf{a}_t \qquad (1)$$

if action $\mathbf{a}_t$ is carried out at time step $t$, and $\mathbf{o}_t$ is the observation at time step $t$.

For simplicity, we often write the embedding $\mathbf{Q}\mathbf{o}_t$ of the current observation as $\mathbf{s}_t$, the internal prediction $\mathbf{Q}\mathbf{o}_t + \mathbf{V}\mathbf{a}_t$ of the embedding $\mathbf{s}_{t+1} = \mathbf{Q}\mathbf{o}_{t+1}$ of the next observation as $\hat{\mathbf{s}}_{t+1}$. The following local synaptic plasticity rules strive to reduce the prediction error:

$$\Delta\mathbf{V}_{t+1} = \eta_v \cdot (\mathbf{s}_{t+1} - \hat{\mathbf{s}}_{t+1})\mathbf{a}_t^T \qquad (2)$$

$$\Delta\mathbf{Q}_{t+1} = \eta_q \cdot (\hat{\mathbf{s}}_{t+1} - \mathbf{s}_{t+1})\mathbf{o}_{t+1}^T, \qquad (3)$$

where $\eta_q$ and $\eta_v$ are learning rates. We refer to Fig. 1b for an illustration. These plasticity rules, commonly referred to as Delta-rules in theoretical neuroscience[18], implement gradient descent towards an adaptation of the observation- and action embeddings $\mathbf{Q}$ and $\mathbf{V}$ with the goal of satisfying Principle I. They represent a form of self-supervised learning since they do not require any target values from an external supervisor. The prediction error $[\mathbf{s}_{t+1} - \hat{\mathbf{s}}_{t+1}] = [\mathbf{Q}\mathbf{o}_{t+1} - (\mathbf{Q}\mathbf{o}_t + \mathbf{V}\mathbf{a}_t)]$ serves as a gating signal for these synaptic plasticity rules, see Fig. S1a in the Supplementary Information. Experimental data from neuroscience that support this type of synaptic plasticity are discussed in section 2 of the Supplementary Information.

The vectors $\mathbf{o}_{t+1}$ and $\mathbf{a}_t$ represent activations of the input layer and the matrices $\mathbf{Q}$ and $\mathbf{V}$ the synaptic weights of the neural circuit depicted in Fig. S1. The gating signals have opposite signs in equ. (2) and equ. (3) because the terms $\mathbf{Q}\mathbf{o}_{t+1}$ and $\mathbf{V}\mathbf{a}_t$ are on

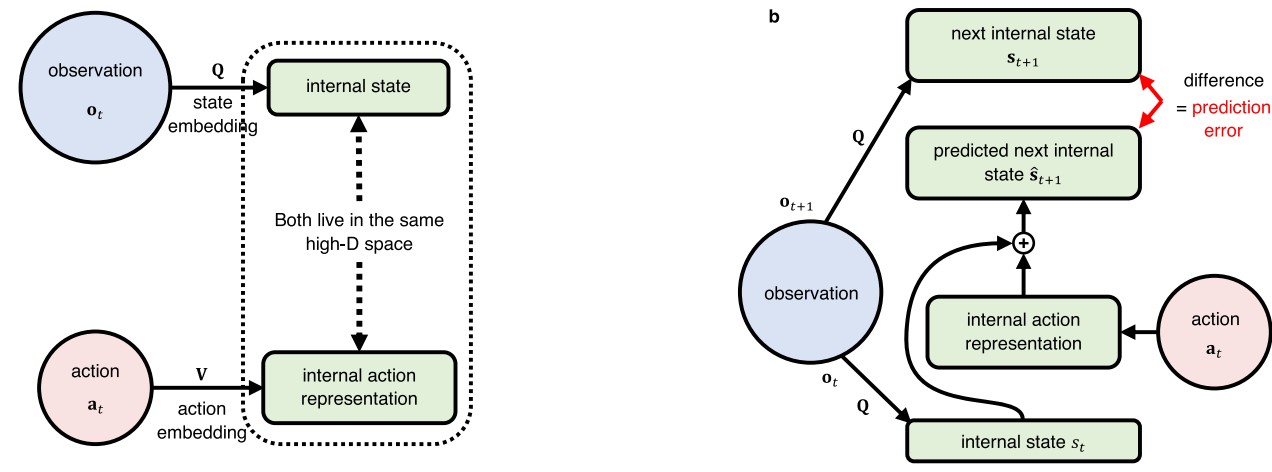

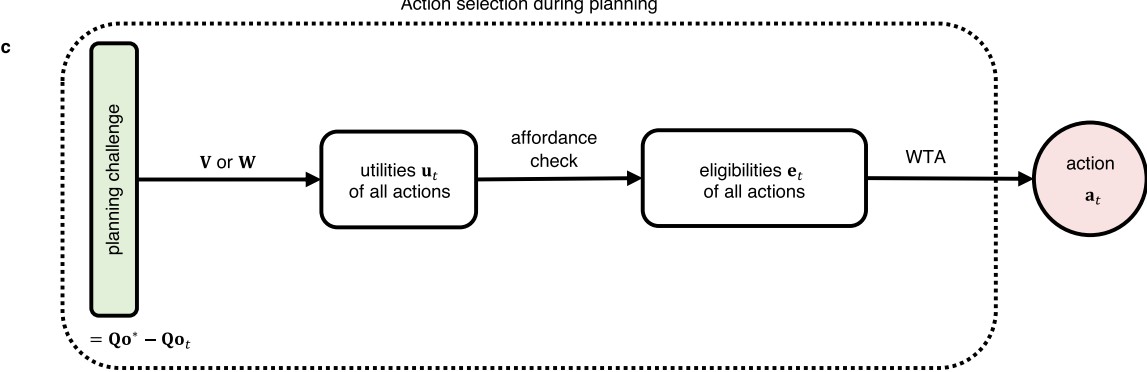

**Fig. 1 | Organization of learning and planning in the CML. a** Both observations from the environment and actions are linearly embedded into a common high-dimensional space S. **b** The learning goal of the CML, see Principle I, is to reduce its prediction error for the next state $s_{t+1}$, i.e., the difference between the prediction $\hat{s}_{t+1} = s_t + Va_t$ and $s_{t+1}$. Prediction errors are reduced by modifying the elements of the embedding matrices $Q$ and $V$, which are represented by synaptic weights in a neural network implementation of the CM (see Fig. S1). **c** The entries of the learned matrix $V$ can be used to generate for the difference between a given target observation $o^*$ and the current observation $o_t$ and any possible action $a_t$ an estimate of its utility, provided the weights between action- and state-representations can be assumed to be symmetric (see Fig. S1). If this assumption is not met, the CML can learn the weights $W$ of a linear map from state representations to action representations by a separate simple learning process, see Fig. S1c (this approach is used in all our demonstrations). Among those actions that can be executed in the current state (which is determined by an affordance check), the one with the largest utility is selected as the next action $a_t$ by a Winner-Take-All (WTA) operation.

opposite sides of equ. (1). Recent experimental data show that there exist so-called error neurons in the neocortex whose firing represents such prediction errors, with both positive and negative signs[19].

Self-supervised online learning with these plasticity rules during exploration can be complemented by letting them also run during replay of episodic memories in subsequent offline phases, as observed in the brain[20]. One can also keep them active during subsequent planning. Making the learning rate $\eta_v$ for $V$ larger than the learning rate $\eta_q$ for $Q$ supports fast adaptation to new actions.

Numerous benefits of predictive coding, a conjectured key feature of brain function, have already been elucidated[21–23]. We show here that it also facilitates planning. More precisely, the plasticity rules (equ. (2) and (3)) automatically also strive to satisfy the following geometric planning heuristic:

**Principle 2.** (An estimate of the utility of an action)

$$u_t(a) = (Qo^* - Qo_t)^T(Va). \tag{4}$$

An estimate of the utility $u_t(a)$ of an action $a$ for reaching some given goal $Qo^*$ from the current state $Qo_t$ is provided by the scalar product.

These utility estimates are reminiscent of value estimates in reinforcement learning[24]. But in contrast to value estimates, they do not depend on a policy and are simultaneously available for every possible goal.

Principle II is directly implied by Principle I if the embeddings of different actions give rise to approximately orthogonal vectors in S. This is immediately clear for the special case where the target observation $o^*$ is the observation $o_{t+1}$ that arises after executing an action $a_t$. Then the term $(Qo_{t+1} - Qo_t)$ is approximately equal to $Va_t$ according to Principle I. Hence, its scalar product with $Va_t$ is substantially larger than with vectors $Va$ that result from embeddings of other actions $a$, provided that all action embeddings have about the same length (see section 3 of the Supplementary Information for a discussion of this issue). Importantly, Principle II remains valid when a sequence of several actions is needed to reach a given target observation $o^*$ from the current observation $o_t$. The difference between the embeddings of $o^*$ and $o_t$ can then be approximated according to Principle I by the sum of embeddings of all actions in a sequence that leads from observation $o_t$ to observation $o^*$. Hence the embeddings of these actions will have a large scalar product with the vector $Qo^* - Qo_t$. Embeddings of actions that do not occur in this sequence will have an almost vanishing scalar product with this vector, provided that action embeddings are

approximately orthogonal. We will often use the abbreviation $\mathbf{Qo}^* - \mathbf{Qo}_t = \mathbf{s}^* - \mathbf{s}_t = \mathbf{d}_t$.

Satisfying the orthogonality condition, at least approximately, requires that the space S is sufficiently high-dimensional. In fact, randomly chosen vectors are with high probability approximately orthogonal in high-dimensional spaces. Therefore we initialize the action embedding matrix $\mathbf{V}$ as a random matrix with entries chosen from a Gaussian distribution, and encode actions $\mathbf{a}$ by one-hot vectors. Then the embeddings of different actions are defined by the columns of the matrix $\mathbf{V}$, and these columns start out to be approximately orthogonal to each other. It will be shown in the next section (see Fig. 2c) that this orthogonality is largely preserved during learning.

Another benefit of a high-dimensional state space is that, when synaptic weights are initialized with random numbers from a Gaussian distribution, the column vectors of the matrix $\mathbf{V}$ tend to have approximately the same length. Furthermore, this approximate normalization will largely be preserved during learning if the state dimension is sufficiently large (see section 3 of the Supplementary Information). A functional benefit of this implicit normalization of the columns of $\mathbf{V}$ is that for the standard case where each action is represented by a one-hot binary vector, each action will be mapped by the embeddings matrix $\mathbf{V}$ onto a vector $\mathbf{Va}$ in the state space that has almost the same length. This entails that the value of the utility $\mathbf{u}_t$, computed according to equ. (4) through a scalar product, depends almost exclusively on the direction of $\mathbf{Va}$, and not on its length.

The implicit approximate normalization of weight vectors, in combination with the approximate orthogonality of action embeddings, entails a functionally important property of the geometry of the learnt cognitive map: The Euclidean distance between the embeddings of any two observations indicates the number of actions that are needed to move from one to the other. More precisely, since embeddings of different actions are approximately orthogonal, the length of the sum of any $k$ action embeddings grows according to the law of Pythagoras with the square root of the sum of squares along the shortest path. Hence, Principle I induces through the iterated application a functionally useful long-distance metric on the cognitive map.

The implicit normalization of weight vectors is best satisfied when the dimension of the state space S is sufficiently large, e.g., in the range of a few 1000 for the concrete applications that are discussed below. Since the dimension of the state space is defined by the number of neurons in a neural network implementation, using a state space with such a dimension provides no problem for brains or for neuromorphic hardware.

If Principle II is satisfied, choosing the next action in order to reach a given goal state $\mathbf{Qo}^*$ can be reduced to a very simple heuristic principle: In first approximation, one just has to choose an action $\mathbf{a}$ whose embeddings points the most into the direction of the goal state. On a closer look, one also needs to take into account that not every action can actually be executed in every state. One commonly refers to the possibility of an action in a given state as its affordance[25]. Particular brain structures, such as, for example the prefrontal cortex, contribute to estimates of these affordances and inhibit inappropriate actions. Since it would be fatal to try out each action in each state, affordance values arise in the brain from a combination of nature and nurture. We assume for simplicity that an outside module provides at any time $t$ a binary masking vector $\mathbf{g}_t$ that denotes for each action whether it is currently available. Multiplication with the vector of utilities yields the vector $\mathbf{e}_t$ of eligibility values for all actions

$$\mathbf{e}_t = \mathbf{u}_t \odot \mathbf{g}_t, \qquad (5)$$

where the $\odot$ operator denotes element-wise multiplication. The CML then selects the action $\mathbf{a}_t$ that currently has the highest eligibility, i.e., it applies a WTA (Winner-Take-All) operation to the vector of eligibility

values and selects an action $\mathbf{a}_t$ with maximal eligibility:

$$\mathbf{a}_t = \mathrm{WTA}(\mathbf{e}_t), \qquad (6)$$

see Fig. 1c for an illustration.

For learning of the CML one just needs to make sure that the synaptic plasticity rules are applied for a sufficiently large number of triples < current state, current action, next state >. It does not matter whether the first two components of these triples are generated randomly, result from randomly generated exploration trajectories, or during the first planning applications. This freedom in the design of the learning phase results from the fact that the CML does not need to remember specific trajectories in order to plan. Rather, it encodes the information that it extracts from each < current state, current action, next state > into its static cognitive map. In the most general case where the CML creates a cognitive map for a general graph, each pair < current state, current action > needs to occur during learning since no inferences can be drawn between their outcomes. In physical realizations of a planning scenario, either for navigation in a physical space of controlling a robot, it turns out that the CML does not need to encounter all possible pairs < current state, current action > during learning because it is able to seamlessly combine learnt knowledge about existing connections with inferences about them based on inherent symmetries of the physical environment.

## Applying the CML to generic planning and problem-solving tasks

Generic problem solving tasks can be formalized as a task to find a path from a given start to a given node in an abstract graph[3]. The directed edges of the graph represent possible actions, and its nodes represent possible intermediate states or goals of the problem-solving process. Each node gives rise to a specific observation in the CML terminology. We consider in this section the general case where no insight into the structure of the graph is provided a-priori, so that all of its properties have to be learnt. We first consider the case of a random graph and then some other graphs that pose particular challenges for online planning.

We test the planning capability of the CML by asking it to select actions that lead from a randomly selected start node to a randomly selected goal node. We evaluate the planning performance of the CML by comparing the length of the resulting path with the shortest possible path, that can be computed by the Dijkstra algorithm. Note that this comparison is a bit unfair to the CML, since the Dijkstra algorithm and other search methods that are commonly used in AI, see[3], receive complete knowledge of the graph without any effort, whereas the CML has to build its own model of the graph through incremental learning. Furthermore, the Dijkstra algorithm and A* are offline planning methods that compute a complete path to the goal, whereas CMLs are online planners according to the definition of[2,3]. In fact, a CML chooses the next action in real-time, i.e., within a fixed computation time that does not depend on the size of the graph, the number of paths from which it has to choose, or how far the target node is away. It also does not need extra computing time when an action becomes unavailable, or if the goal suddenly changes. In contrast, Dijkstra's algorithm and A* need to compute a complete trajectory from the current node to the goal before they can produce the next action. In addition, A* has to run again from scratch if the start or goal node changes. Dijkstra's algorithm has to do that also if the start node changes.

**Generic random graphs.** We focus first on the example of a randomly connected graph with 32 nodes, where every node has a random number between two and five edges to other nodes. The edges of the graph are undirected since they can be traversed in any direction. But each traversal of an edge in a given direction is viewed as a separate action, hence there are two actions for every edge. The random graph

**a**

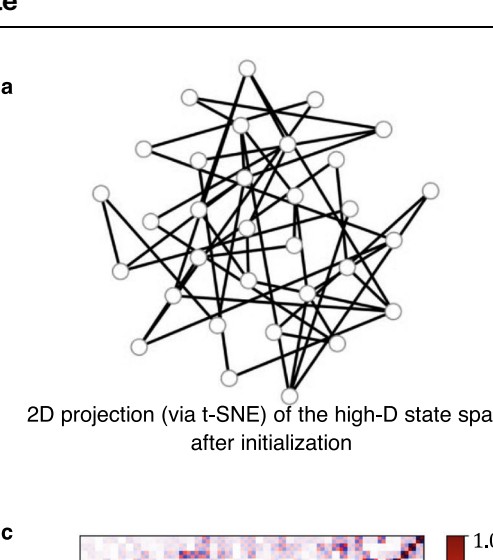

2D projection (via t-SNE) of the high-D state space
after initialization

**b**

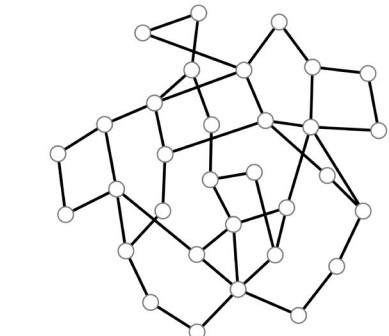

2D projection (via t-SNE) of the high-D cognitive map
created by the CML

**c**

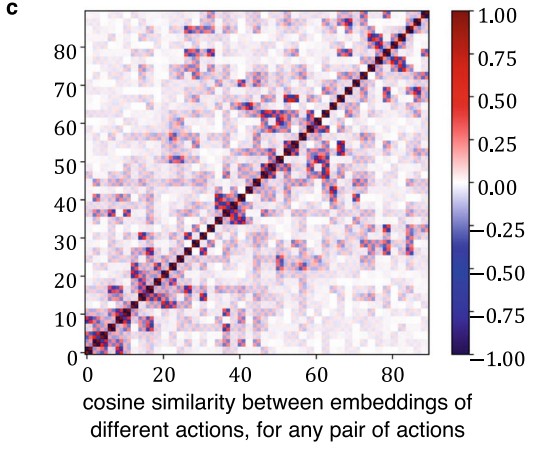

cosine similarity between embeddings of
different actions, for any pair of actions

**d**

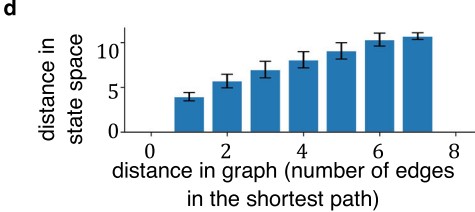

**e**

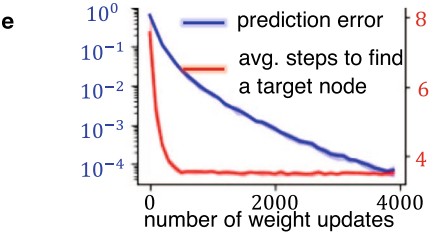

**f**

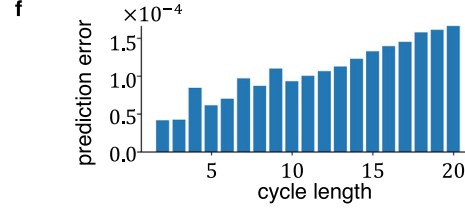

**g**

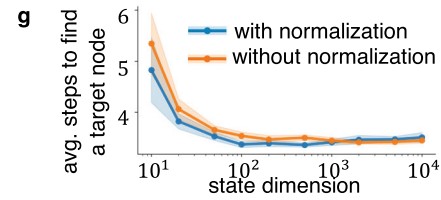

**h**

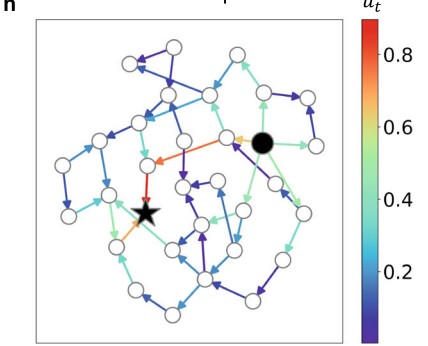

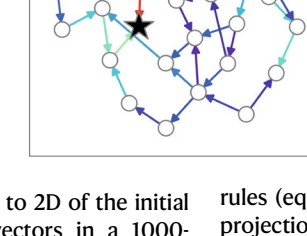

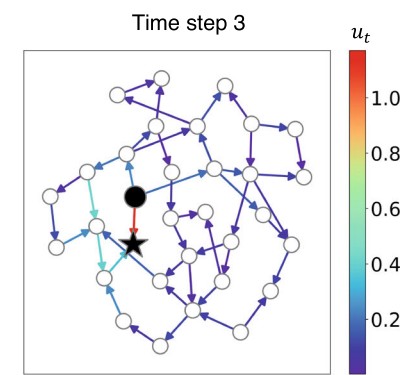

is visualized in Fig. 2a through a t-SNE projection to 2D of the initial representation of its states through random vectors in a 1000-dimensional space.

We use a very simple exploration strategy during learning: The CML takes 200 random walks of length 32 from randomly chosen starting points. These random walks are subsequently replayed while learning of the embeddings **Q** and **V** through the synaptic plasticity rules (equ. (2) and (3)) continues. Fig. 2b shows a corresponding 2D projection of the states which the CML has constructed during this learning process. Most pairs of nodes that are connected by an edge are now the same distance from each other. This results from the fact that the CML has learnt to approximate their difference by the embedding of the action that corresponds to the traversal of this edge. One clearly sees from the 2D projection of the learnt cognitive map in

**Fig. 2 | Properties of the cognitive maps that the CML builds through self-supervised learning. a** 2D-projection (via t-SNE) of the initial state embedding of a generic random graph. **b** 2D projection of the state embedding for the same graph after learning. The emergent geometry of the cognitive map supports planning a path from any start node to any goal node through Principle II. **c** Cosine similarity of embeddings of different actions under the learned embedding V. Most action embeddings are orthogonal to each other, with some exceptions for actions that start at the same node (these are adjacent in the chosen order). **d** The average distance between any pair of nodes in the learned cognitive map can be approximated by the square root of the length of the shortest path that connects them. Error bars indicate two standard deviations of the distribution of distances in state space for a given distance in the abstract graph. **e** Learning curves of the CML for the given random graph. As the prediction error decreases (blue curve), the online planning by the CML rapidly learns to find near-optimal paths to given goals (red curve). The successful planning does not require perfect error minimization, which shows CML's robustness. The shaded backgrounds indicate two standard deviations of the variables across the 10 training rounds. **f** When one applies Principle I iteratively to the edges of any cycle in the graph, the sum of the embeddings of the corresponding actions should be close to the 0-vector. One sees that this is satisfied quite well for the given random graph. **g** The CML's performance improves as the state dimension increases and plateaus when the state dimension is sufficiently large. The explicit normalization on each column of **V** (see equ. S2) is beneficial for CML at smaller dimensions, but it becomes unnecessary for higher-state dimensions. The shaded backgrounds indicate two standard deviations of the variables across the 10 training rounds. **h** Example for a given start node (black disc) and goal node (black star) in the random graph. Utilities of possible actions are indicated for each step during the execution of a plan, provided they have positive values. One sees that utilities are online adjusted when the current node (black ball) moves. Furthermore, several alternative options are automatically offered at most steps during the execution of the plan.

Fig. 2b that it enables a very simple heuristic method for moving from any start to any goal node: Just pick an action that roughly points into the direction of the goal node. Hence even path finding in a random graph that has no geometric structure is reduced by the CML to a simple geometric problem: One just has to apply Principle II.

Producing a cognitive map with a geometry that approximately satisfies Principles I and II, requires solving a quite difficult constraint satisfaction problem. Principle II requires that action embeddings remain approximately orthogonal. Fig. 2c and d show that this is largely satisfied for the random graph that we consider. Panel d demonstrates in particular that the distance between any two nodes can be approximated by the square root of the number of actions on the shortest path that connects them, as expected from the law of Pythagoras if the embeddings of actions on this shortest path have unit length and are orthogonal to each other. Principle I requires in addition that the embedding of any sequence of actions that form a cycle in the random graph sum up to the 0-vector. Fig. 2f shows that this constraint is also approximately satisfied by the cognitive map which the CML has generated through its local synaptic plasticity. But it is obviously impossible to satisfy both constraints in a precise manner: If one just considers cycles of length 2, where one moves through an edge in both directions, each represented by a different action, one sees that these two actions cannot be orthogonal if they sum up to the 0-vector.

CML effectively addresses this challenge by intentionally breaking orthogonality when necessary. Initially, high-dimensional vectors are naturally orthogonal to one another. However, as the learning progresses, it necessitates the breaking of orthogonality for actions that move the agent to similar/opposite node groups to be positively/negatively correlated. This deviation doesn't impact planning, as such changes effectively encode the relationship between actions in a cognitive map.

Figure 2h gives an example for the continual adjustment of utility estimates for each action while a plan for reaching the given goal node is executed. No learning takes place between these steps, but the estimates of utilities are adjusted when the starting node moves. Note that typically several alternative choices for the next action have high utility during the execution of this plan. This reduces the reliance on a single optimal path, and provides inherent flexibility to the online planning planner.

The CML produces for this graph a trajectory from an arbitrary start node to an arbitrarily given target node using an average of 3.480 actions (std. deviation 1.538). This is very close to the average shortest possible path of 3.401 (std. deviation 1.458), which the Dijkstra algorithm produces.

**Finding the least costly or most rewarded solution of a problem.** Different actions incur different costs in many real-world problem-solving tasks, and one wants to find the least costly path to a goal, rather than the shortest path. The cost of an action could represent for

example the effort or time that it requires, or how much energy it consumes. Not only the human brain is able to provide heuristic solutions to these more challenging problem solving tasks, but also non-human primates[26]. Hence the question arises of how neural networks of the brain can solve these problems. One can formalize them by adding weights (=costs) to the edges of a graph and searching the graph for the least costly path to a given goal, where the cost of a path is the sum of the costs of the edges that it traverses.

A simple extension of the CML provides heuristic solutions for this type of problems. One can either let the cost of an action regulate the length of its embedding via a modification of the normalization of columns of **V** according to equ. 5, or one can integrate its cost into the affordance values $\mathbf{g}_t$ that are produced by the affordance module. The impact on action selection seems to be the same in both cases, since the eligibility of an action is according to equ. 11 the product of its current affordance and utility. But in the case of the first option, the cost of an action will affect the geometry of the whole cognitive map, and thereby support foresight in planning that avoids future costly actions. Integrating cost values into the affordances has the advantage that they can be changed on the fly without requiring new learning. But costs affect only the local action selection in a greedy manner. We demonstrate in an example that this simple method supports already planning in weighted graphs very well. We assigned random integer weights (= costs) between 4 and 7 to the edges of the previously considered random graph. Corresponding affordance values $\mathbf{g}_t$ are shown in Fig. 3b for the cognitive map which the CML produces after learning. The CML produces for random choices of start and goal nodes in this graph a solution with an average cost of 18.76. This value is quite close to optimal: The least costly paths that the offline Dijkstra algorithm produces have an average cost of 18.00.

Also, many reinforcement learning problems can be formulated as path-planning problems in a weighted graph. In this case the inverse of the weight of an edge encodes the value of the reward that one receives when this edge is traversed. Hence the same CML heuristic as before produces a path from a given start to a given goal that accrues an approximately maximal sum of rewards. In many reinforcement learning tasks one does not have a single goal node, but instead a set of target nodes in which the path from the start node should end. An example is the graph shown in Fig. 3c. Note that this graph does not have to be a tree. It encodes the frequently considered case of a 2-step decision process[2,3]. The task is here to find a path from the bottom node to a node on layer 2 that accrues the largest sum of rewards. This is equivalent to finding the most rewarded path to a virtual node that is added above them, with added edges from all desirable end nodes. Hence also this type of task can be formalized as the task to find the least costly path from a given start to a given goal node in a weighted graph. Therefore the CML provides approximate solutions also to such multi-step decision processes that are commonly posed as challenges for reinforcement learning[3]. Note also that the type of reinforcement

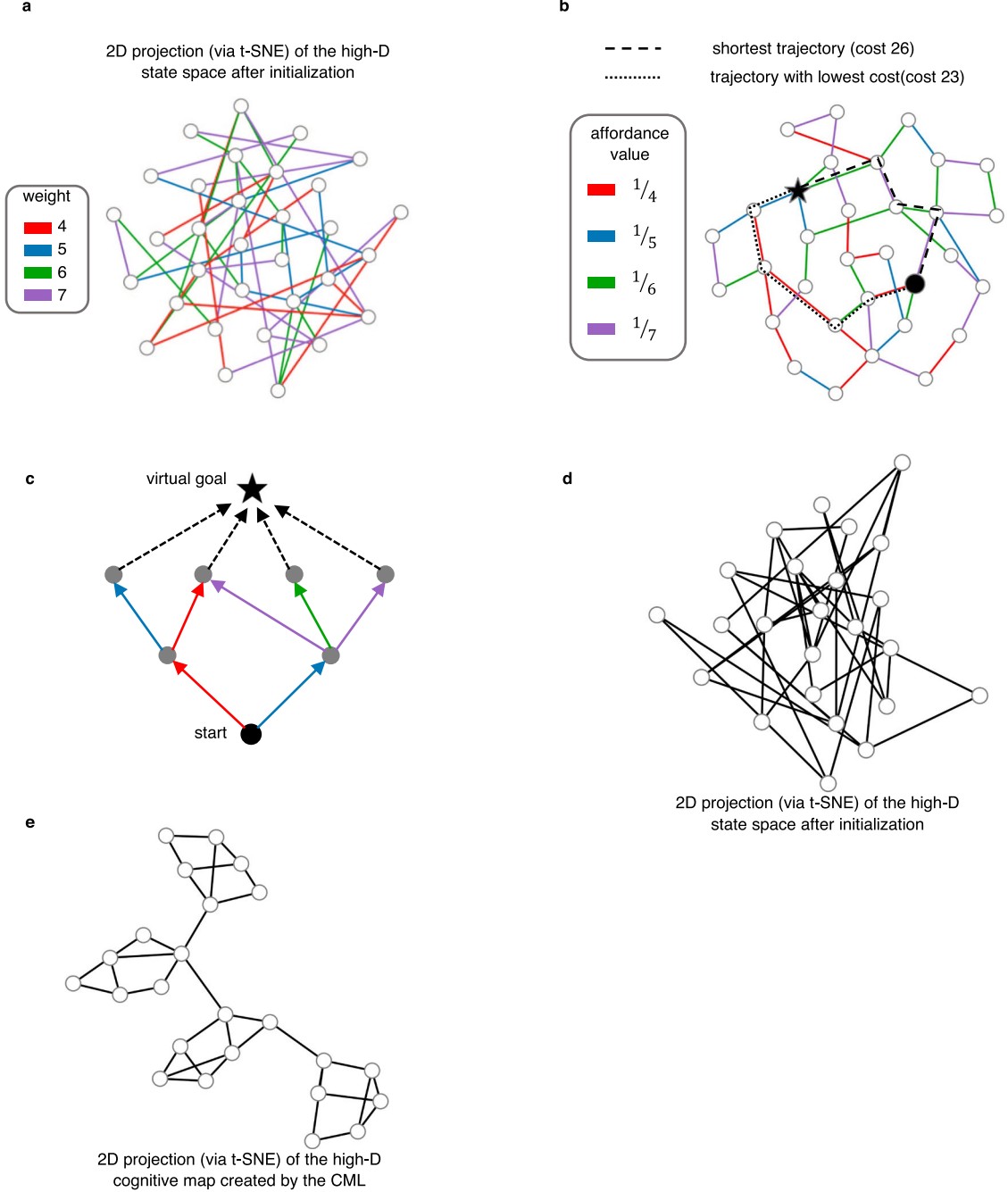

**Fig. 3 | More demanding problem solving challenges for the CML.** The weights in panel **a** were sampled uniformly from a set of integers containing 4, 5, 6, and 7. The affordance values in panel **b** were obtained by dividing 1 by the weight of the corresponding edge. This makes the CML prefer low-cost actions. Panel **c** illustrates a simple method for reducing a commonly considered class of reinforcement learning problems, 2-step decision processes, to a task where a path to a given goal node has to be found: One adds a new virtual goal node that receives edges from all terminal nodes of the decision process. Weights, or rather their inverse, can

encode, in this case, the rewards that are given when an edge is traversed. This injects a bias to prefer rewarded actions. Panel **d** Small-world graphs, like the one whose 2D-projection of its initial state space (via t-SNE) is shown here, could pose particular challenges to a heuristic planner, especially if a particular node has to be visited to escape a local cluster. However, one sees in panel **e** that the learnt cognitive map makes it possible to find the way out of each cluster through a simple geometric heuristic: go into the direction of the goal.

learning problems that were tackled with the elegant flexible method of ref. 27 have a fixed set of terminal states. Since there only the transition to a terminal state yields a reward, the task to find a maximally rewarded solution is exactly equivalent to finding a least costly path to an added virtual node which receives weighted edges from all terminal states. Hence these problems can also be solved by the CML. An interesting difference is that the CML does not need to have a default policy, and requires only local synaptic plasticity for learning.

**Graphs that pose special challenges for online planning.** Small world graphs can be seen as special challenges, since one needs to identify the precise node that allows escaping from a local cluster. Therefore we tested the CML on a graph that had four clusters each consisting of 6 densely interconnected nodes, but only a single connection from one cluster to the next. The initial state embedding is shown for this graph in Fig. 2d. The cognitive map which the CML produced for this graph after learning is depicted in Fig. 2e. The CML

solved all planning tasks for given pairs of start and goal nodes very efficiently. It produced in each of 1000 samples of such tasks the near optimal solution, with an average path length of 3.651 (std. deviation 1.981), which is very close to the average shortest possible path of 3.644 (std. deviation 1.977).

Another concern is that online planning gets confused when multiple paths of the same length lead to the goal, since this appears to require some knowledge about the global structure of the graph. But this feature did not impede the performance of the CML for the example graphs that we examined: The CML needed in the graph that is shown in Fig. S3a on average 3.374 (std. deviation 1.557) actions to reach the goal (average over 1000 trials with randomly selected start and goal nodes). This value is optimal according to the Dijkstra algorithm. In other words, the CML found in each trial a the shortest possible path. A sample trial is shown in Fig. S3a.

A further concern is that a learning-based online planner may have problems if there are dead ends in a graph: Paths into these dead ends and paths for backing out of them are traversed during learning, and could potentially become engraved into the cognitive map and cause detours when it is applied to planning. To explore how the CML copes with this difficulty, we designed a graph which contains a large number of dead ends. The graph as well as the utility values computed by the CML during the execution of a plan can be seen in Fig. S3b. We found that despite the difficulty of having to deal with dead ends, the CML managed to achieve the same performance as Dijkstra on this graph.

## Path planning in physical environments

If actions are state-invariant, such as the action *move left*, symmetries that exist in the underlying physical environment could enable predictions of the impact of an action in a state without ever having tried this action in this state. In fact, it is known that the cognitive map of rodents enables them to carry out this type of generalization: Rodents integrate short-cuts into their path planning in 2D mazes which they had never encountered during exploration[28]. Hence a key question is whether the cognitive maps which the CML learns can also support this powerful form of generalization. Therefore we are considering from now on only CMLs with state-independent, i.e., agent-centric, codes for actions. In general not all actions can be executed in each state, and information about that is provided to the CML by an external affordance module. We tested this in a task that is inspired by the task design in[29] and[30]. In the simplest case the environment is a rectilinear 2D grid of nodes, where each node gives rise to a unique observation which is denoted by an icon in Fig. 4 (we did not label different observations by numbers or letters because this might suggest a linear order of the observations, which does not exist). At each node there are the same 4 actions A, B, C, and D, available, see Fig. 4a, each of them encoded by the same one-hot code at each node. The CML has initially no information about the meaning of these actions. But the observations that were provided after an action resulted in an interpretation of each of them as a step in one of the 4 cardinal directions of the grid. The only information that the CML received during learning were the observations, see Fig. 4b for examples. We allowed 22 exploration sequences, all of length 3. We made sure that not all edges of the graph were encountered during learning, as indicated in Fig. 4c. No additional information about the geometry of the environment was provided to the CML, and it had to infer the structure of the 2D maze from accidentally encountering the same observation in different contexts, i.e. in different sequences of observations. We tested the CML after learning, especially for start and goal nodes that never occurred both on any exploration sequence, so that the CML had to recombine knowledge from different exploration sequences. Furthermore, we tested it on planning challenges where the shortest path contained edges that were never encountered during learning, see Fig. 4d and e for an example. Nevertheless, the CML produced solutions for these

planning tasks that had the shortest possible length, and made use of unexplored edges, see Fig. 4e for an example.

The cognitive map which the CML had generated during learning, see Fig. 5a, explains why it was able to do that: The PCA analysis of this high-dimensional cognitive map shows that the CML had morphed it into a perfect 2D map of spatial relations between the observations that it had encountered during learning. An animation illustrating the emergence of its cognitive map during the learning phase can be found in Supplementary Movie 1.

One sees that the internal representations of observations keep changing until a perfectly symmetric 2D map has been generated. This self-organization is reminiscent of earlier work on self-organized maps[31]. But there the 2D structure of the resulting map resulted from the 2D connectivity structure of the population of neurons that generated this map. In contrast, there is no such bias in the architecture of the CML. Consequently, the same CML can also create cognitive maps for physical environments with other symmetries, e.g. for 3D environments. Since cognitive maps for 3D environments are hard to visualize, we show instead in Figs. S4 and S5 an application to a hexagonal 2D environment, where the same generalization and abstraction capability can be observed as for the rectilinear environment.

In Fig. 5b we show the result of applying to the cognitive map which the CML has generated an abstract measure for the compositional generalization capability of neural representations: the parallelism score[32]. The very high parallelism score for the two state differences that are examined there indicates that a linear classifier can discern for any pair of observations which of them is attained from the other by applying a particular action, even for pairs that were not in the training set for the linear classifier.

## Goal-oriented quadruped locomotion emerges as generalization from self-supervised learning

We wondered whether the CML can also solve quite different types of problems, such as control of simulated robots, through generalization of experience from self-supervised learning. We tested this on a standard benchmark task for motor control[33,34]: Control of simulated quadruped locomotion (see Fig. 6a) by sending torques to the 8 joints of its 4 legs (note that one commonly refers to this motor system as an ant, in spite of the fact that ants have more than 4 legs). Whereas the goal was originally only to move the ant as fast as possible in the direction of the x-axis, we consider here a range of more demanding tasks. Observations were encoded by 29-dimensional vectors $\mathbf{o}_t$, which contained information about the angles of the 8 joints, their respective speed, the (x,y) coordinates of the center of the torso, and the orientation of the ant, see Table 1 for details.

In the learning phase, the CML explored the environment through 300 trajectories with random actions i.e., through motor babbling, see the video at Supplementary Movie 2. The observations during these trajectories were the only source of information that the CML received during learning. In particular, the CML was never trained to move to a goal location. We found that it was nevertheless able to control flexible goal-directed behavior. In the simplest challenge, the CML had to navigate the ant to an arbitrary goal location that was 20 meters away from the starting position (see the video at Supplementary Movie 3). In this case, the target observation $\mathbf{o}^*$ was defined by the desired (x, y) coordinates of the center of the torso, while leaving all other components of the observation the same as in the current observation $\mathbf{o}_t$. The resulting trajectories of the ant are depicted in Fig. 6b, where the stars indicate the goal locations and the line plotted in the corresponding color indicates the path taken by the center of the ant's torso. The CML solved this task with a remaining average distance to the target of less than 0.465 m (average over 1000 trials). Fig. 6c illustrates the correlation between the

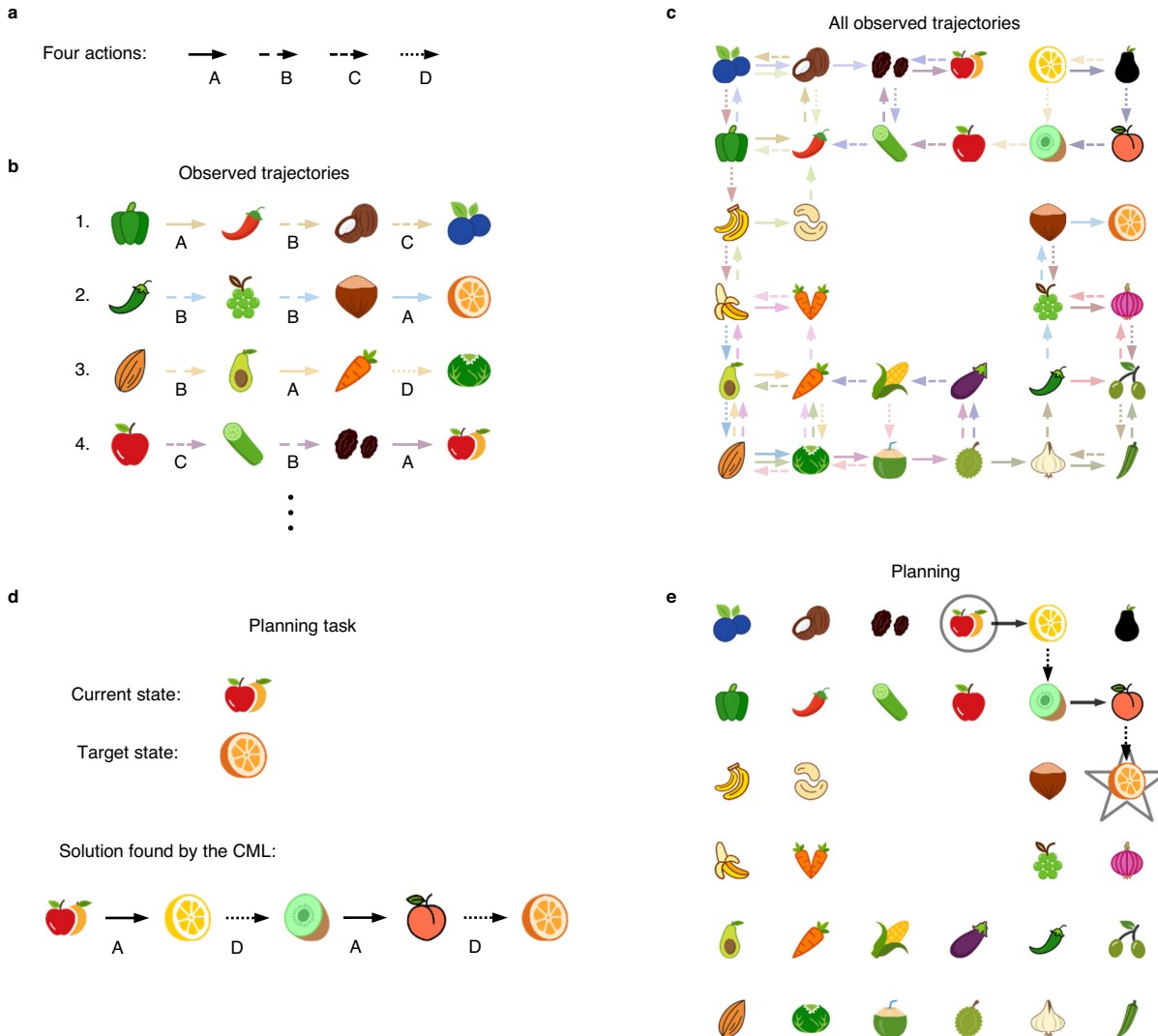

**Fig. 4 | The CML learns fundamental properties of a 2D environment, and exploits them for generalization. a** Four different actions, which can be applied in any state, although the affordance module does not allow their application if they would lead out of the range of the given 2D environment. The CML has no prior knowledge about their meaning in the 2D environment. Panel **b** shows samples of the 22 trajectories that the CML has observed during learning from the perspective of the CML: Only relations between actions and observations are provided in short trajectories with three actions. Panel **c** provides a birds-eye view of all 22 trajectories during learning, Note that some combinations of actions and states were never tried out. Panel **d** provides an example of a planning task. A start observation and a target observation are given to the CML, and it has to produce a sequence of actions which let it move on the shortest path from start to goal. **e** The solution of the task from panel **d**, which the CML produced. Note that three of its four actions had never been tried out from these states (observations) during exploration (compare with panel **c**). Hence the CML is able to generalize learnt knowledge and apply it in new scenarios.

state prediction error and the performance of the CML in terms of the remaining distance to the target location.

Due to the task agnostic nature of CML learning it can solve also completely different tasks without any further learning, such as tasks with continuously changing goals. We considered two such tasks, see Fig. 6d. In one, fleeing from a predator, the first two coordinates of the target observation were defined by a position 5m away from the current ant position in the direction away from the current position of the predator (marked by the center of the red square in Fig. 6d). In another task, mimicking the chasing of prey, the first two coordinates of the target observation $o^*$ were defined by the current position of the center of the red ball while the ball moved around arbitrarily. Also these new tasks were handled very well by the same CML, see the online available videos for fleeing from a predator video at Supplementary Movie 4, or chasing a target video at Supplementary Movie 5.

## Considerations for the implementation of CMLs in neuromorphic hardware

CMLs are well-suited for implementing planning, problem solving, and robot control in several types of energy-efficient neuromorphic hardware: Digital neuromorphic hardware such as SpiNNaker 1[35] from the University of Manchester, SpiNNaker 2[36], Loihi 1[37] and Loihi 2[38] from Intel, the hybrid Tianjic chip from Tsinghua University[39], as well as analog neuromorphic hardware that employs memristor arrays for in-memory computing in discrete time, an approach that is pursued by IBM and several other companies and universities[40–44].

All these neuromorphic chips support the implementation of neurons that produce non-binary output values; hence the linear neurons that are employed by the CML (see Fig. S1) can easily be implemented on them. In-memory computing chips enable an especially fast evaluation of the main computational operation for planning, equ. (4), with just 3 invocations of the crossbar.

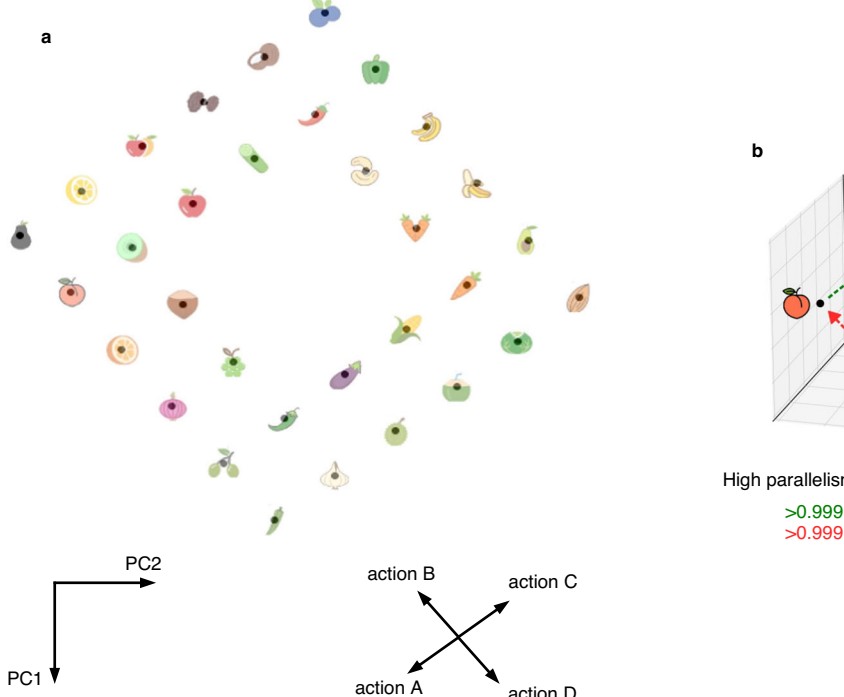

**Fig. 5 | Structure of the cognitive map which the CML generated during exploration of the 2D environment of Fig. 4.** The same incomplete exploration of a 2D environment as in the previous figure was used. **a** Projection of the learnt cognitive map to its first two principal components. Additionally, the columns of V, i.e., the embeddings of the 4 actions, are shown in the same 2D projection. One sees that the learnt relations between actions and observations perfectly represent the ground truth, thereby explaining why the CML produces optimal shortest paths during planning for each given start and goal observation. Panel **b** provides the parallelism score of the learnt cognitive map for pairs of states where one results from the other by applying a specific action (two actions are indicated by red and green arrows), in two different contexts (= initial states). The parallelism score is the cosine between the state differences that result from applying the same action in two different contexts. According to[32] the parallelism score is a useful measure for the compositional generalization capability of learnt internal representations. It is applied there both to the brain and to artificial neural networks.

The WTA operation has commonly been used on numerous neuromorphic chips, and there exists extensive literature on that. For example, it employed on SpiNNaker[45] for solving constraint optimization problems, and on Loihi[46] when implementing the K-nearest neighbor algorithm (its readout mechanism is equivalent to WTA when K=1). WTA can be implemented directly by local processing units or through lateral inhibition with inhibitory neurons according to the model of theory from[47]. Note that the CML does not require high precision for the WTA computation. If several actions have eligibility values close to the maximum, it does in general not matter which one is chosen.

A delay of one time step that is used in the neural network implementation of Fig. S1 for inverting the sign of a signal can be implemented through an inhibitory relay neuron, or more directly through a buffer.

In contrast to backpropagation or backpropagation through time, the local synaptic plasticity rules that the CML employs are suitable for on-chip learning. For chips that employs memristors, the embedding weights **Q**, **V**, and **W** that the CML learns can be implemented as updates of memristor memory. We found that a weight precision of 8 bits suffices for learning to plan in the abstract random graph from Fig. 2 without a significant performance loss. Such 8 bits weights can be implemented directly in the type of memristors that had been presented in[48]. But also memristors with less precision can be employed with the help of the bit-slicing method[49,50].

In contrast to previous methods for problem solving on neuro-morphic chips, one does not have to program the concrete problem into the chip. Rather, CML explores the problem space autonomously and encodes it in a data-structure on the chip, the cognitive map, that enables low-latency solutions for a large array of tasks.

## Relation of the CML to self-attention approaches (Transformers)

CMLs rely like Transformers on self-supervised learning of linear embeddings of data into high-dimensional spaces. But there also exist structural similarities between its basic equations and those of self-attention mechanism described in[12], the work horse behind the Transformer architecture. There one computes:

$$\text{Attention}(\mathbf{Q},\mathbf{K},\mathbf{V}) = \text{softmax}\left(\frac{\mathbf{Q}\mathbf{K}^{T}}{\sqrt{d_k}}\right)\mathbf{V}, \qquad (7)$$

where **Q** corresponds to a matrix containing the queries, **K** to a matrix containing the keys, **V** a matrix containing the values, and $d_k$ is a scaling constant. During self-attention, the matrices **K**, **Q**, and **V** are computed directly from a sequence of tokens. The CML on the other hand does not use a sequence of tokens, it rather only considers a single token, which is the target direction in state-space $\mathbf{d}_t = \mathbf{s}^* - \mathbf{s}_t$ and can be interpreted as a single query. This feature arises from the implicit assumption that observations are Markovian, i.e., only the most recent observation is relevant for making a prediction. As there is no sequence of tokens, unlike the Transformer, the CML employs fixed keys **K** and values **V** that do not depend on the current observation or action.

In the CML, the next state prediction $\hat{\mathbf{s}}_{t+1}$ can be written as:

$$\hat{\mathbf{s}}_{t+1} = \mathbf{s}_t + \text{WTA}(\mathbf{g}_t \odot \mathbf{K} \cdot \mathbf{d}_t)^T \mathbf{V}^T, \qquad (8)$$

where $\mathbf{d}_t = \mathbf{s}^* - \mathbf{s}_t$ contains information about both the current state $\mathbf{s}_t$ and the target state $\mathbf{s}^*$ acts as the query, $\mathbf{g}_t$ represents the affordance gating values, which indicate the availability of an action in the current

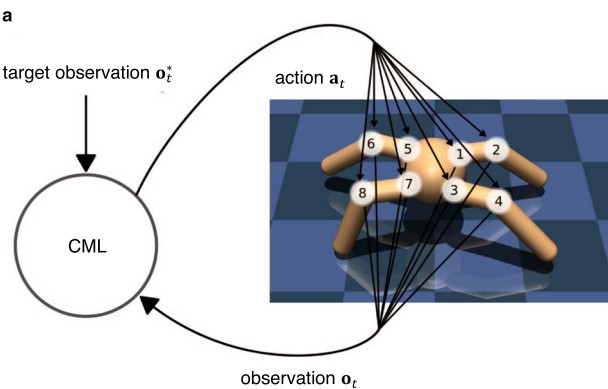

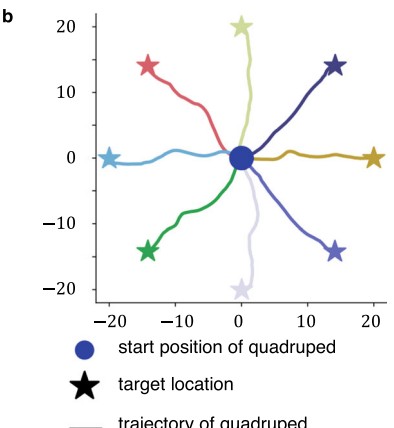

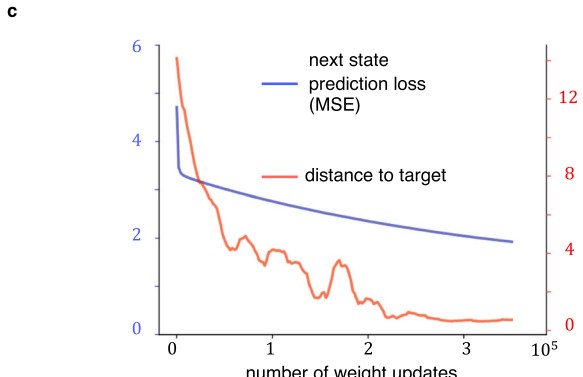

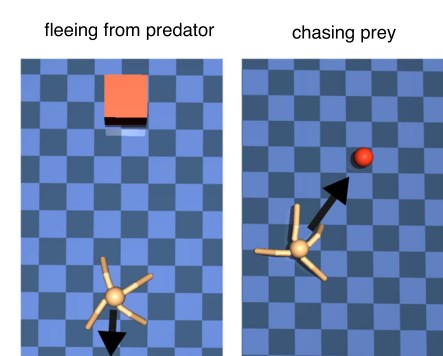

**Fig. 6 | Application of the CML to motor control.** Panel **a** depicts the quadruped (ant) with its 8 joints that can be controlled individually. Panel **b** provides first evidence for the generalization capability of the CML controller. Although it only used random actions (motor babbling) during learning, it was immediately able to move the ant to a given goal. Resulting paths of the center of the ant torso are shown for 8 sample goal states. The blue circle in the middle indicates the starting position of the ant, and the stars depict 8 different goal positions. Trajectories to these goals which were generated by the CML were low-pass filtered and plotted in the same color as the goal. The ant was reset to the center location after every episode. Panel **c** visualizes the evolution of the next state prediction error during learning (blue curve), as well the average distance to the target location after the end of the episode (red curve) when learning was interrupted after every 2000 weight updates the partially learnt cognitive map was applied to the task of panel b, averaging over 32 evaluation trajectories. Panel **d** illustrates two further tasks that can be solved by the same CML without any new learning or instruction: fleeing from a predator(Supplementary Movie 4) and chasing a prey (Supplementary Movie 5).

position in the environment and the keys **K** are considered to be equal to the values **V** in the CML. Equation (8) displays striking structural similarities with the attention mechanism used in Transformers (equ. (7)), especially with respect to the relationships among the variables involved, if one takes the previously described differences into account. The latter entails that the attention matrix is not quadratic of dimension $\mathbb{R}^{(n_{seq}, n_{seq})}$, where $n_{seq}$ is the sequence length of the input tokens, as in Transformers. Rather, the attention matrix in the CML corresponds to the utilities $\mathbf{u}_t$ and has the shape $\mathbb{R}^{(1, n_{seq})}$, as there is only a single query $\mathbf{d}_t$. Note that $n_{seq}$ corresponds to the number of actions $n_a$ in this analogy. In the CML, the attention matrix measures the utility for every action, assigning a high utility value to actions that are more useful and a lower value for less useful actions. Therefore, one could say that the CML *attends* to useful actions.

It is also noteworthy, that while the attention mechanism in Transformers uses a softmax function, the CML uses a WTA function instead. However, these two functions are also familiar, as the softmax function is, with increasing input values, in the limit the same as the WTA function. We also want to point out that the addition of the current state $\mathbf{s}_t$ in equ. (8) is reminiscent of residual connections, that are also widely used in Transformer architectures. This is a common practice in the literature for neural network-based approximations of Transformers, see[51–53].

## Discussion

The capability to plan a sequence of actions, and to instantaneously adjust the plan in the face of unforeseen circumstances are hallmarks of general intelligence. But it is largely unknown how the brain solves such tasks[2]. Online planning methods that are both flexible, i.e., can be employed for reaching different goals, and highly performant are also missing in AI[3]. We have shown that a simple neural network model, that we call a cognitive map learner (CML), can acquire these capabilities through self-supervised learning to predict the next sensory input. We have demonstrated that for a diverse range of benchmark tasks: General problem-solving tasks, formalized in terms of navigation on abstract graphs (Figs. 2, 3), navigation in partially explored physical environments (Figs. 4, 5), and quadruped locomotion (Fig. 6). Surprisingly, the CML approximates for navigation in abstract graphs the performance of the best offline planning tools from AI, such as the Dijkstra- and A*-algorithms, although it plans in a much more flexible and economical online manner that enables it to accommodate ad-hoc changes of the goal or in the environment.

A fairly large class of problem-solving tasks can be formalized as a task to find a shortest path to a given goal in an abstract graph, see[3]. Hence the CML can be seen as a problem solver for such tasks. But the CML cannot solve constraint satisfaction problems such as the Traveling Salesman Problem, for which neuromorphic-hardware-friendly

**Table 1 | Components of an observation $o_t$ for the case of the ant**

| index | Description |
|---|---|
| 1 | x-coordinate of the torso |
| 2 | y-coordinate of the torso |
| 3 | z-coordinate of the torso |
| 4 | x-orientation of the torso (quaternions) |
| 5 | y-orientation of the torso (quaternions) |
| 6 | z-orientation of the torso (quaternions) |
| 7 | w-orientation of the torso (quaternions) |
| 8 | angle between torso and first link on front left |
| 9 | angle between the two links on the front left |
| 10 | angle between torso and first link on front right |
| 11 | angle between the two links on the front right |
| 12 | angle between torso and first link on back left |
| 13 | angle between the two links on the back left |
| 14 | angle between torso and first link on back right |
| 15 | angle between the two links on the back right |
| 16 | x-coordinate velocity of the torso |
| 17 | y-coordinate velocity of the torso |
| 18 | z-coordinate velocity of the torso |
| 19 | x-coordinate angular velocity of the torso |
| 20 | y-coordinate angular velocity of the torso |
| 21 | z-coordinate angular velocity of the torso |
| 22 | angular velocity of angle between torso and front left link |
| 23 | angular velocity of the angle between front left links |
| 24 | angular velocity of angle between torso and front right link |
| 25 | angular velocity of the angle between front right links |
| 26 | angular velocity of angle between torso and back left link |
| 27 | angular velocity of the angle between back left links |
| 28 | angular velocity of angle between torso and back right link |
| 29 | angular velocity of the angle between back right links |

methods had been described in[54]. On the other hand the CML does not require that the problem to be solved is hand-coded in the neural network. Rather, the CML explores autonomously the problem space, and can also plan action sequences for reaching new goals that did not occur during learning.

The CML makes essential use of inherent properties of vectors in high-dimensional spaces, such as the fact that random vectors tend to be orthogonal. It adds to prior work on vector symbolic architectures[13–15] a method for learning useful high-dimensional representations.

Learning to predict future observations, which is the heart of the CML, has already frequently been proposed as a key principle of learning in the brain[21–23]. But it had not yet been noticed that is also enables problem solving and flexible planning.

The CML provides a principled explanation for a set of puzzling results in neuroscience: High-dimensional neural codes of the brain have been found to change continuously on longer time scales, while task performance remains stable[55–58]. The combination of these seemingly contradictory features is an automatic byproduct of learning a cognitive map: Neural codes for previously experienced observations need to be continuously adjusted in order to capture their relation to new observations. But this does not reduce the planning capability of the neural network as long as some basic geometric relations between these neural codes are preserved, see movie at Supplementary Movie 1 for a demonstration.

We have also shown that the CML shares some features with Transformer, in particular self-supervised learning of relations

between observations as the primary learning engine, and the encoding of learnt knowledge in high-dimensional vectors. But the application domain of CMLs goes beyond that of Transformers since they support self-supervised learning of an active agent. Another important difference is that the CML requires only Hebbian synaptic plasticity, rather than backpropagation of error gradients. Also, its small sets of weights enable the CML to learn from small datasets. An attractive goal for future research is to combine CMLs with Transformers in order to combine learning of the consequences of own actions with enhanced learning from passive observations. Also hierarchical models are of interest that plan and observe on several levels of abstraction, see[59] for biological data and[60] for an extension of the CML in this direction.

Models for learning to choose actions that lead to a given goal have so far been mainly based on the conceptual framework of Reinforcement Learning (RL). But the learning processes of the most frequently discussed RL methods aim at maximizing rewards for a particular reward policy, e.g., for reaching a particular goal. Hence they provide less insight for understanding how the brain attains the capability to respond to new challenges in a flexible manner. They also do not support flexible robot learning. The key concept of most RL approaches is the value function for states, and most RL approaches aim at first learning a suitable value function.

In the endotaxis approach of[61] the agent learns in fact value function for several different potential goals. These value functions can subsequently be employed for planning paths to any of these pre-selected goals through a neural circuit, provided that each edge of the underlying graph is viewed as a separate action, like in our CML applications to abstract graphs, but unlike our other CML applications. Hence the agent cannot employ during planning any edges of the graph that it has not yet traversed.

It is worth noting that a value function is a fundamentally different data structure than the cognitive map that a CML learns, because this cognitive map does not depend on particular goals. Among RL methods that aim at alleviating this dependence we would like to mention three. In model-based RL one also learns a goal-independent model of the environment. But whereas the cognitive map that is learnt by the CML can be used instantly for producing actions that lead to any given goal, model-based RL needs to employ a rather time-consuming computation such as policy- or value iteration for that.

In another important RL approach one first learns a successor function, that predicts future state occupancy for any number of steps[62,63]. The successor function can then be used to compute efficiently the value function for any new reward policy. But the successor function itself depends on a particular exploration policy, and therefore provides suboptimal state predictions when a new goal requires a policy that significantly differs from the one that was used during exploration. In contrast, the cognitive map that the CML learns is largely independent of the policy that is applied during learning, it only depends on the set of <state, action> pairs that are encountered during learning. Furthermore, the primary advantage of the successor function approach is that it supports fast estimates of a value function for a given goal. But like for model-based RL, one still needs to employ a rather time-consuming computation such as policy- or value iteration in order to produce actions that are likely to lead to a given goal, and we are not aware of proposals for neuromorphic implementations of that.

In linear reinforcement learning[27] the computation of an optimal policy to a new goal is greatly simplified. It is in that sense that RL approach that is the most similar to the CML approach, although we are not aware of proposed neural network implementations of it. The linear reinforcement learning approach is based on the assumption that one can a-priori specify a default policy for subsequent behavior, and that it is adequate to penalize deviations from this default policy that are used to reach a particular goal. In contrast, the CML does not require an assumption about a default policy. We are not aware of

neuromorphic implementations of any of the RL approaches that we have discussed here. We also want to point out that in contrast to most RL approaches, CML learning only requires local synaptic plasticity rules that are easy to implement in digital energy-efficient hardware and memristor-based analog hardware for in-memory computing. Hence CMLs are likely to support new designs and implementations of robot control where control commands for flexible behavior are computed by a neuromorphic chip with low latency and low energy-consumption.

## Methods

### Mathematical description

The observation $\mathbf{o}_t$ is embedded into the state-space of the CML using the embedding matrix $\mathbf{Q} \in \mathbb{R}^{n_s, n_o}$:

$$\mathbf{s}_t = \mathbf{Q} \cdot \mathbf{o}_t, \tag{9}$$

where $n_s$ is the dimensionality of the state space and $n_o$ the dimensionality of the observation $\mathbf{o}_t$. This also holds for the target state: $\mathbf{s}^* = \mathbf{Q}\mathbf{o}^*$.

In accordance with Principle I, the next state prediction of the CML can be written as:

$$\hat{\mathbf{s}}_{t+1} = \mathbf{s}_t + \mathbf{V}\mathbf{a}_t. \tag{10}$$

During planning one first computes the current utility values for all actions:

$$\mathbf{u}_t = \mathbf{V}^T \mathbf{d}_t, \tag{11}$$

where $\mathbf{d}_t$ is the vector pointing from the current state to the target state $\mathbf{s}^* - \mathbf{s}_t$. This equation is a paralleled process of Principle II: Each dimension of the resulting vector $\mathbf{u}_t$ represents the utility value of one corresponding action. In case the transpose is not convenient to calculate, one can use the learnt $\mathbf{W}$ (according to equ. S1) to substitute $\mathbf{V}^T$:

$$\mathbf{u}_t = \mathbf{W}\mathbf{d}_t. \tag{12}$$

To account for scenarios where not all actions are applicable in each state, or where different actions incur different costs, the vector of utility values for all actions is element-wise multiplied with the vector of affordance values $\mathbf{g}_t$ to yield the vector of eligibilities for all actions, see equ. (5).

To select an action, the action with the highest eligibility is selected by applying the WTA function to the vector of eligibility values of all possible actions, see equ. (6).

### Details to planning on abstract graphs

For planning on an abstract graph every node is encoded by a unique one-hot encoded observation $\mathbf{o}_t$. Every action $\mathbf{a}_t$ corresponds to traversing an edge of the graph in a certain direction. Hence, there are two actions for every edge of the graph, which are all encoded by one-hot vectors. To explore the graphs, the CMLs were allowed to traverse the graph, taking 200 random trajectories through the environment where each trajectory was of length 32 steps. These two numbers determine the size of the training set. Generally, the size should be large enough to ensure that every distinct action is explored at least once.

As a baseline comparison, the Dijkstra graph search algorithm (see Alg. 1) was employed. It is noteworthy that in this task only graphs with equally weighted edges are considered, and therefore $A^*$ search is equivalent to Dijkstra, as there are also no heuristics that can be applied to the type of graph that we consider.

The initial values for $\mathbf{Q}$, $\mathbf{V}$ and $\mathbf{W}$ were drawn from a Gaussian distribution: with ($\mu = 0, \sigma = 1$) for $\mathbf{Q}$, and ($\mu = 0, \sigma = 0.1$) for $\mathbf{V}$ and $\mathbf{W}$.

The planning performance depends very little on these parameters. Initializing $\mathbf{V}$ with smaller values improves the CML's performance. One can also initialize $\mathbf{V}$ by all zeros and the performance is about the same. The dimensionality of the state space of the CML was 1000 for the abstract graph tasks, which was selected based on Fig. 2g.

The learning rates were $\eta_q = 0.1$, $\eta_v = \eta_w = 0.01$ on all abstract graph tasks. The CML algorithm is not sensitive to the precise values of the learning rate. We made the learning rates for $\mathbf{V}$ and $\mathbf{W}$ by an order of magnitude smaller than for $\mathbf{Q}$ because they were initialized by an order of magnitude smaller. In general we found that good results an be achieved with any learning rates in the range of [0.005, 0.5].

All variations of planning in abstract graphs that we discuss— Random, Small World, Dead End, and Multi-path graphs (see Fig. S3)— employed the same parameters: The same initialization parameters ($\mu$ and $\sigma$) for $\mathbf{Q}$, $\mathbf{V}$ and $\mathbf{W}$, the same learning rates $\eta_q$, $\eta_v$, and $\eta_w$, and the same state dimension. In fact, the same parameters can also be used for all navigation tasks in physical environments that we consider. This shows that the performance of the CML is not very sensitive to these parameters. For the quadruped control task we used a larger state dimension and smaller learning rates. These parameters could also be used for all the other tasks, but the smaller learning rate would unnecessarily increase their training times.

In all tasks the function $f_a$ was chosen to be a Winner-Take-All (WTA) function, which takes a vector as an argument and returns a vector indicating the position of the highest valued element of the input vector with a 1 and setting all other elements to 0, see equ. (13).

$$\mathrm{WTA}(\mathbf{x}) = \mathbf{v}, \quad \text{where } \mathbf{v}_i = \begin{cases} 1 & \mathbf{x}_i = \max(\mathbf{x}) \\ 0 & \text{else}. \end{cases} \tag{13}$$

Note, in equ. (13) if $\exists\, i, j, i \neq j$: 7D1$\mathbf{v}_i = \mathbf{v}_j = \max(\mathbf{x})$, then the WTA function would return a one-hot encoded vector indicating index with the lowest value.

The vector $\mathbf{g}_t \in \mathbb{R}^{n_a}$ of affordance values is defined for the case of unweighted graphs as a k-hot encoded vector, whose components with 1 indicate that the corresponding action can be executed in the current state (observation), i.e., corresponds to an outgoing edge from the current node in the graph.

Details to weighted graphs: The affordance values for weighted graphs were computed by dividing 1 by the weight of the corresponding edge in the graph. This way, selecting actions with large weights (costs) is discouraged. To avoid that the CML gets stuck in a loop it turned out to be useful to disallow the selection of the same action twice within a trial.

### Details to 2D navigation

Two different environment geometries were considered in this task: rectangular and hexagonal (see Fig. S4) environments. Each observation $\mathbf{o}_t$ was one-hot encoded. An action was interpreted in the environment as a move from one cell in the grid to a neighboring cell. Hence there were four actions in the rectangular environment while there were six actions in the hexagonal environment. In the environments of the 2D navigation tasks all actions were executable (affordable) in every state, as long as the action would not cause the move outside of the boundary of the grid. Consequently, the affordance values in $\mathbf{g}_t$ had value 1 for all actions unless the action would result in moving outside of the grid, in which case it had value 0. We assume that these affordance values are provided by some external module. In principle one could also learn them, but this would make the model substantially more complex.

One can use here the same parameters as for the abstract graph tasks. However, a state space dimension of 80 suffices here (it can even be chosen smaller without affecting the outcome). The planning performance of the CML was robust to changes of parameter values also for the navigation tasks in physical environments. Successful learning

can also be achieved with a state space dimension of 2 or higher, and other learning rates below 1 also worked. We set all learning rates to be 0.5 in all 2D navigation tasks.

The parallelism score was obtained by computing the cosine similarity between the state differences of the observations indicated in Fig. 5b. The state difference between two observations $\mathbf{o}^a$ and $\mathbf{o}^b$ can be computed by simply embedding the observations into state space and then computing the difference: $\mathbf{Q}\mathbf{o}^b - \mathbf{Q}\mathbf{o}^a$.

**Details to controlling a quadruped**

**Details to relating action commands $a_t$ to actions in the environment.** An action remapping scheme is used to remap a one-hot encoded action resulting from the WTA to a dense action consisting of 8 torque values that can be used to control the ant model. For this mapping, only torques of strength $\pm 0.1$ were considered. As there were 8 controllable joints, this yields a set of $2^8 = 256$ different combinations of considered torques. Each of these combinations was considered as one possible action. These combinations were enumerated in a list and the one-hot encoded action $\mathbf{a}_t$ was used as an index for selecting a combination of torques in this list. Furthermore, each action was applied 10 times in the environment to guarantee a larger change in the environment.

Given its complexity, this task required a larger state space dimension of 4000 and smaller learning rates $\eta_q = 0.0025$ and $\eta_v = 0.0005$. One could use these parameter settings also for the other tasks, but that would slow down learning for them. We used here the learnt matrix $\mathbf{V}$ for generating estimates of the utility. The ant task receives continuous valued observation, instead of one-hot observations as the other tasks. Therefore it required a somewhat different weight initialization: The initial values for the two matrices $\mathbf{Q}$ and $\mathbf{V}$ were drawn from a Gaussian distribution, with $\mu = 0$ and $\sigma = 0.1$, where for $\mathbf{V}$, $\sigma = 1$.

**Details to the observations of the ant.** A detailed table explaining the 29 dimensional vector of the observation $\mathbf{o}_t$ can be found in Tab. 1.

**Details to the use of target observations for the ant.** A key feature of the CML is that every type of task is formulated as a navigation problem to a target state $\mathbf{s}^*$ in state space, where $\mathbf{s}^*$ results from embedding a desired target observation $\mathbf{o}^*$ using $\mathbf{Q}$. Consequently, it is possible to design a desired target observation $\mathbf{o}^*$, which can be passed to the CML, which then in turn tries to find a sequence of actions with the intent of receiving a current observation $\mathbf{o}_t$ from the environment which matches the target observation $\mathbf{o}^*$. The resulting flexibility is underlined by the three different types of problems presented in ant controller: moving to a target location, fleeing from an adversary and chasing a target. For each of these tasks, different target observations $\mathbf{o}^*$ are computed and passed to the CML. All three tasks are based on navigation and therefore use the first two fields in the observation (see Tab. 1) to set a desired target location using Cartesian coordinates. In the moving to target location task, the target location is chosen to be fixed and has an initial distance of 20 meters away from the ant. In chasing a target task, the target location is moving and is therefore updated every time step. In the fleeing from an adversary task, the target location is updated every time step as well and set to 5 meters away from the ant, in the direction pointing directly away from the adversary. Furthermore, the target location is passed to the ant relative to itself and not the absolute location as defined by the global coordinate system. To transform the absolute location to a target location relative to the ant first a vector $\mathbf{v}_{at}$ which points from the ant to the target location is computed. This vector can be written in polar coordinates, where $|\mathbf{v}_{at}|$ is the magnitude and $\phi_{at}$ corresponds to the angle. To correct for the error that would occur if the ant itself is rotated, we deduct the angle of the ant itself (to the x-axis) $\phi_a$ from $\phi_{at}$. Finally, the target coordinates can be computed by adding the polar vector with the angle $\phi_{at} - \phi_a$ and magnitude $|\mathbf{v}_{at}|$ to the current position of the ant. As can be seen in Tab. 1 the observation also consists of 27 other fields in addition to the coordinates. The remaining fields of $\mathbf{o}_t^*$ are chosen to be equal to $\mathbf{o}^t$, as this induces the CML to only change the current position of the center of its torso, and not any other variable from $\mathbf{o}_t$.

**Details to the regularization of Q.** During the learning process, the CML might try to focus on predicting some parts of the next observation $\mathbf{o}_t$ which are easily predictable in order to minimize the prediction error. An example for components of the observation $\mathbf{o}_t$ that are easy to predict are the angles of the joints, while the spatial position of the ant is harder to predict. A problem can therefore arise when the CML adapts the embedding $\mathbf{Q}$ so that the spatial position has little impact on the vector in the state space onto which the observation vector is mapped. To ensure that variables that are important for planning are well-represented in the state space, $\mathbf{Q}^T$ was regularized to reconstruct the observation $\mathbf{o}_t$ in this task using a similar learning rule as for all other learning processes. This also induces the CML to work with state representations that are very informative about the observations to which they correspond.

**Details to the computation of $g_t$.** Actions can entail one of two possible local actions per joint, which apply torques in one of the two directions in which the joint can rotate. Furthermore, every joint has two limits, which represent the minimum and maximum angles that the joint can have. The affordance of an action depends on the ant controller on the current angle of each joint. If a joint is already close to one of its limits, the action bringing it even closer to the limit receives a low affordance value (close to 0), and the action moving it away from the limit receives a high affordance value (close to 1). As the CML considers the actions to be one-hot encoded, the affordance values $\mathbf{g}_t$ are computed by averaging over the individual affordance values that result from this heuristics for each joint.

**Details to the sampling of actions.** To allow the CML to explore the environment, actions are sampled randomly. The sampling process takes the bounds of the angles into account, selecting torque values which are unlikely to move the joint too close to the bound with a higher probability.

## Data availability
Our work does not utilize any specific dataset, making it independent of proprietary data sources.

## Code availability
An example code for the CML algorithm on all abstract graph tasks (Random, Small World, Dead End, and Multi-path graphs), is available at https://github.com/IGITUGraz/Cognitive-Map-Learner.

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

## Acknowledgements

We would like to thank Anton Arkhipov, Guozhang Chen, Denis Kleyko, Oleg Kolner, Thomas Ortner, Philipp Plank, Danielle Rager, Christopher Summerfield, Yujie Wu, and Joshua Yang for advice and helpful comments on an earlier version of this paper. The research of WM was partially supported by the Human Brain Project of the European Union (Grant Agreement number 945539), a grant from Intel, and the National Science Foundation of the USA (EFRI BRAID project 2318152). Supported by TU Graz Open Access Publishing Fund.

## Author contributions

C.S. conceived the algorithmic approach. C.S. and W.M. together developed the conceptual and mathematical framework for it, and designed the experiments and plots in the paper. Y.Y. showed that explicit weight normalization is not needed during learning if one works in a sufficiently high-dimensional state space. The technical execution and experimental validation of the study were primarily driven by the joint efforts of C.S. and Y.Y. The paper was primarily written by W.M., except for Methods and the Supplementary Information, which were primarily written by C.S., with substantial contributions from Y.Y. The manuscript was collectively refined and finalized by all authors.

## Competing interests

The authors declare no competing interests.
