## [Peer Review File · Nature Communications]

REVIEWER COMMENTS

Reviewer #1 (Remarks to the Author):

This is a refreshing article that addresses learning in neural networks that does not attempt yet again to approximate back-propagation with local learning rules. The authors address the challenge of developing biologically plausible mechanisms for learning to produce sequence of actions and/or to plan them. The results are noteworthy from a modeling perspective. However the claims that the methods can be implemented in energy-efficient neuromorphic hardware are not sufficiently detailed.

The authors do not make sufficient comparisons to relevant or related literature. Therefore it is difficult to assess how the proposed approach compares to alternative ones. For example, a paper on predictive learning that is closely related to the work proposed, but not mentioned is here:

<<https://www.nature.com/articles/s41467-021-21696-1>>

The authors should explain how their approach compares to the work mentioned above and show where it differs and how it improves that previous approach.

More generally, the claim that there is a lack of compelling models for brain inspired sequence learning and problem solving is not totally correct, as there are several models that in one way or another have addressed (although partially) such challenge. For this aspect, the authors should explain how their work differs, or extends previously published models. I am providing examples of papers that are related to this, but please keep in mind that this is not an exhaustive set, and the authors should perform a thorough literature search to give credit to previous models and highlight the differences/advantages/disadvantages:

- <https://doi.org/10.1162/NECO_a_00893>

- <<https://doi.org/10.1162/neco.2009.11-08-901>>

- <<https://doi.org/10.1523/JNEUROSCI.4098-12.2013>>

- <<https://doi.org/10.3389/neuro.10.023.2009>>

- <<https://doi.org/10.1371/journal.pcbi.1010233>>

The authors do mention that models of goal-directed action sequences have been indeed proposed in the Reinforcement Learning literature, but they do not explain what the differences are with respect to the approach proposed. It would be useful to have a review of the models that are closest to the

presented one and to clarify their advantages/disadvantages. The authors should also compare their model with related work on neural network approaches to Constraint Satisfaction Problem solving, including the neural sampling framework of the same group (but, e.g., see also <https://doi.org/10.3389/frai.2020.580607>).

In addition there have been several publications on neuromorphic systems for these types of problem-solving tasks that the authors should also consider:

- <https://doi.org/10.1073/pnas.1212083110>

- <https://www.nature.com/articles/ncomms9941>

- https://doi.org/10.1162/NECO_a_00785

Based on these comparisons they should revise their claims, that nothing exists yet (already in the abstract) all the way up to the discussion (e.g., lines 528-530).

As the claims are that the model proposed is inspired by real neural circuits, can be used to solve planning tasks in physical environments and be applied to robotics and real-time neuromorphic hardware, the authors should formulate the model and equations in continuous time rather than discrete time. This will unveil potential problems related to causality already starting from equations (2) and (3), e.g., when are the various signals at $(t+1)$ actually produced? which one arrives first? what happens between instants (t) and $(t+1)$? how can signals at (t) be multiplied by signals at $(t+1)$? how are these signals kept around? what are the required time-constants of the system? how do they relate to the dynamics of the input signals? what are the delays (if any) needed to support the model? This is a critical point that could potentially invalidate the whole model.

In addition to the critical discrete-continuous time issue, it would be helpful, just for sake of clarity, to have a neural network diagram with the connectivity between the units representing the observation signals, the action signals, etc. The authors make a strong point (already from the title) about their model using a local synaptic plasticity rule that is compatible with biology and neuromorphic hardware friendly. But for other parts of the model it is not clear if this locality principle applies. For example the weight normalization of eq. 5 appears to require knowledge of all values of a whole column. The authors should clarify if this corresponds to all inputs afferent to a neuron (e.g., referring to the neural network diagram figure requested above) or to all neuron activations. This would allow them to discuss about biologically plausible mechanisms for achieving such normalization (e.g., homeostatic plasticity, synaptic scaling, etc.), but it would also force the authors to discuss about the time scales and dynamics of this process (e.g., compared to the dynamics of learning) which are crucial for both biology and hardware implementations. Another global mechanism is the WTA operation of eq. (7). The authors should explain why/how this does not invalidate the advantages of the local learning mechanism used. Similarly, they should further clarify if/how the element-wise multiplication of eq. (6) can be done in parallel, locally without requiring sequential CPU-like processing. To their benefit, the authors do mention in-memory computing and cross-bar arrays of memristors that could support this operation in principle, but then

they claim that their model could be mapped onto the Loihi neuromorphic platform, which does not use such cross-bars. To be really convincing that the model can be implemented on energy-efficient neuromorphic hardware the authors should be more precise and substantiate these claims with detailed specifications. On a side note, both the Loihi platform and the SpiNNaKer platform mentioned in the manuscript were not designed to minimize energy consumption (e.g. the Loihi chip has 2 standard 8086 CPUs that burn the standard CPU power consumption). So in that respect the authors should not refer to them as "energy efficient" neuromorphic systems.

In addition to comparing to the classical computer science algorithms as benchmark (e.g., Dijkstra's algorithm) the authors should compare performance with other neural network approaches (e.g., see literature mentioned above, but also the e-prop work from the same group). Benchmark comparisons with specific accuracy numbers have become the standard (for better or worse) in recent literature and they should be added here as well. Some choices made by the authors are not properly justified (e.g., why 200 and 32 at lines 222-223?), and some details are missing (e.g., "after a while" on caption of Fig. 2e). The caption of Fig.2f does not explain what the figure is about. It is more a comment or an observation on the data. It is not clear from Fig.2f if the error goes all the way to zero and would stay at zero if learning were to continue (i.e., if the network stays in the global minimum once it finds it or if it can jump out of it later).

Section at line 263 should relate the problem presented to the "Traveling Salesman Problem", and at the end of that part it would be interesting to see if the model proposed could be used to solve generic SAT problems, to compare with standard benchmarks used in the CSP literature.

The section starting at line 389 should clarify that the robotic demonstration is *virtual* and no real robot was used in this work. The authors should make it clear from the title that it is a *simulated* *virtual* experiment.

The section on Neuromorphic Hardware should be substantiated with more details (see comments above), for example explaining how to map the model (how many neurons, how many synapses per neuron, how many connections, how to implement the WTA, etc.).

Typical neuromorphic hardware platforms (and even ANN accelerators) try to reduce the precision of the weights to 4 or 3 bits. So the requirement to use more than 6 bits severely restricts the possibility to implement such model on generic neuromorphic hardware and limits it to only the current generations of SpiNNaKer and Loihi (as future optimized ones will likely reduce the weight bit resolution).

The section on the relation to transformers is tangential to the work proposed and (in an effort to reduce size) could be moved to a supplementary material part.

The discussion should highlight even more (perhaps bringing that argument further up) how the model proposed requires "only" Hebbian plasticity and does not need back-propagation.

There is not sufficient details to understand how to reproduce the results. The authors should provide a more comprehensive description in the methods and provide access to the code.

Reviewer #2 (Remarks to the Author):

Review of Stöckl and Maass (2023) “Local prediction-learning in high-dimensional spaces enables neural networks to plan”

The authors propose a neuromorphic algorithm by which an agent can learn the structure of an environment and decide on the best course of action to reach a given target location in the shortest time. The principle is to embed both the states (graph nodes) and the actions (graph edges) in the same high-dimensional space, such that the graph distances are represented by geometric distances between nodes, and the actions correspond to directions in the space. Then the agent can pursue the shortest route by choosing the action that points in the direction of the target node. The embeddings of both states and actions are learned during random exploration of the graph, based on optimizing a prediction of the next state from the previous state and action. Ultimately the embedding should be such that the next state can be predicted by simply adding the action vector to the previous state. The authors argue that all this can be accomplished with biologically plausible learning rules.

Evaluation

This is an interesting suggestion for how an agent might learn the structure of its environment, namely how states and actions are connected. This ultimately results in a goal signal, by which the correct action can be chosen that leads to the goal in the shortest amount of time. Some of my questions concern whether the algorithms are really neuromorphic, in the sense that they could be implemented in biological neural circuits.

Concerns about bio-plausibility:

1. The learning rule for state-action association, Eqns 2 and 3: This looks difficult to implement with neurons, because:

a) it involves two different time points on the post-synaptic side: s_{t+1} appears at time $t+1$, but \hat{s}_{t+1} at time t ;

b) to make s_{t+1} appear in that neural population, the action vector must temporarily be switched off, requiring some kind of control system;

c) the postsynaptic variable is a difference between two successive time points. It is not obvious how to implement such a delta rule with biophysics.

2. The computation of utility, Eqn 4: Again, how would this be implemented with neurons?

a) It involves multiplying a population vector with another vector in the same population; what would be the mechanism for such a multiply-and-add operation?

b) In the left term, the activity is driven by an action; in the right term, activity of the same neurons is driven by observations. How would that be accomplished?

c) The right term involves subtracting activity from the current observations from activity driven by some remembered observation. How are the remembered observations stored? Are these fed into the network one after another? How are they subtracted?

3. Normalization of synaptic weights

a) Eqn 5: Here each input synapse gets modified by a factor that depends on the strengths of all the other synapses onto that same neuron. This violates the locality of synaptic plasticity: A synapse should be modified based only on its own strength and the activity of the pre-and postsynaptic neurons.

b) Another related normalization appears in line 660.

4. Winner-take-all choice (Eqn 7): Explain how this would work in a neuromorphic system? How does the agent sample the values of the affordable actions? By actually executing them in the real world one at a time and comparing the resulting output from Eqn 4? But then the observations o_t would change as all the actions are played out. Or is the comparison done “mentally” without real-world action? If so, does this require some accessory system that can look up the codes of all the affordable actions and feed them into the network one at a time?

In summary, if the authors advocate that this model could be implemented in brains, it would be helpful if the proposal included a bio-plausible neural circuit for each of these operations.

Other questions:

5. Graphs with cycles

a) As pointed out (p.9), the theory behind this learning model causes problems with cyclic graphs. All the graph edges are supposed to be orthogonal in the embedding space, yet adding the edges around a loop should give zero. Both can't be true. It is not clear how this conflict was resolved. Does it require careful parameter tuning so as to sustain performance of the model?

b) Line 585 describes a hack by which the agent was prevented from traveling cycles during exploration. That requires oracular knowledge of the graph. How would that work in a real-world system?

6. The concept of "action"

a) In the first part of the paper (p.1-12), every edge on the graph is considered a different action, and they are all encoded with one-hot vectors. For example, a robot action like "turn left" would have a different one-hot code at every different location. This is somewhat different from the common concept of "action", which is specific to the agent's movements, not where they are executed. By postulating a one-hot code for every edge on the graph one effectively circumvents the problem of path integration, which requires making a prediction from stringing together actions that may appear identical from the agent's perspective.

b) Please explain how a naive agent entering a new environment will already have a one-hot code for every edge on the graph, even if they involve identical movements of the agent.

c) What if the actions instead were agent-centered, e.g. {left, right, forward, back} as in the second part of the paper (p.13ff). Would the system still be able to learn an arbitrary graph?

7. "Affordances"

a) What is the source of the "affordances" in Eqn 6. They represent part of the graph structure. Don't these need to be learned as well? In the present formulation it seems they are offered to the system without learning (contra line 194).

b) By what mechanism would this happen in a neuromorphic system, and how would they be represented by the neural circuit?

c) Figure 4e highlights the fact that the agent takes an action that was never taken during exploration (Fig 4c). But what if there was a wall between those two nodes, and that's why the action never occurred there? How does the agent know that the action is "affordable".

8. Parameter sensitivity

a) It appears that for each of the illustrated graphs the embedding space had a different dimensionality, and different choices were made for the learning rates (p.22). In line 610, the state space has fewer dimensions than the number of actions, which obviously precludes finding orthogonal vectors. What motivated all these different choices?

b) How robust is the system to the relevant parameters? Can the same agent with one parameter set learn different spaces? The report would benefit from exploring a wider range of graphs and scanning over parameter values.

9. Related work

There is prior research on these topics that could be used to put the current work in context. In particular, various versions of model-based reinforcement learning acquire the structure of the state space through learning, so that a goal-directed policy can be superposed on that. The present paper only deals with learning the state space, not with learning the goal locations: Those are provided by some unspecified accessory system. Within the RL literature there has been recent enthusiasm about the "successor representation", which is an embedding that helps predict the agent's next state. For neuromorphic models that learn the successor representation, see for example Fang 2023 (<https://doi.org/10.7554/eLife.80680>) and literature cited there.

Other suggestions:

10. There is frequent reference to "planning". But the model presented here doesn't make any plans. Once it arrives at a node it decides on the next action, as though it had a lookup table (see line 13). It does not "think ahead". In the neuroscience and robotics field, "planning" usually concerns evaluating the outcome of successive actions ahead of time, for example comparing the value of different routes. The behavior implemented here is more like "online navigation". This may be confusing to the reader.

11. In Figures 4 and 5, is there a meaning to the emojis painted on the nodes? At least this reader finds them distracting. Preceding figures worked just fine without emojis.

12. In part 2 (p.13ff) the main difference from part 1 is that the actions are now agent-centric. It may help the reader to point this out. Because there are only 4 action vectors, the predictions are forced into a 4-dimensional subspace of the high-dimensional space. Eventually, the learning process squeezes that into a 2-dimensional subspace that accurately reproduces the geometry of the graph.

13. Eqn 4: should start with $u_t =$

14. Figure 6c: Perhaps show a bit more of the time course to document that the two variables have settled.

15. Fig 2 caption: “square root of the length of the shortest path” should this be “square root of the sum of squares along the shortest path”. As stated, it doesn’t resemble the law of Pythagoras.

16. Fig 4a: use different arrows for the 4 actions.

17. Typos:

Line 68

Line 32

Line 250: Meaning of this sentence unclear.

Line 392

Line 505

Eqn 12: should be V_a instead of V ?

29.11 2023

Rebuttal for NCOMMS-23-33922

We are grateful for the critical and constructive remarks from both reviewers. Their thorough work has made it possible to substantially improve our approach and its presentation. In particular, we have redone all experiments, added new figures and text which address their concerns, and edited the text throughout in order to avoid misunderstandings. We are also providing the code for the applications of CML to the general planning and problem solving tasks.

Reviewer #1 (Remarks to the Author):

- This is a refreshing article that addresses learning in neural networks that does not attempt yet again to approximate back-propagation with local learning rules. The authors address the challenge of developing biologically plausible mechanisms for learning to produce sequence of actions and/or to plan them. The results are noteworthy from a modeling perspective. However the claims that the methods can be implemented in energy-efficient neuromorphic hardware are not sufficiently detailed.

We have added in Fig. S1 a possible neural network implementation of the CML, and we have explained this construction in the new section 1 of the Supplement. Furthermore, we have made the section on possible neuromorphic hardware implementations on p 21 – 22 substantially more specific.

- The authors do not make sufficient comparisons to relevant or related literature.

We have made another effort to identify all relevant literature, and to compare our approach with related ones. We could not find any methods in the literature that could solve the same tasks as the CML. Perhaps the reviewer thinks that approaches based on learning and replaying sequences can solve the same tasks as our approach, since all of the literature which the reviewer suggests are based on such methods. But with such methods an agent could not produce close-to-shortest plans for reaching any given goal node from any given start node in a general graph (see our first demo), solve planning challenges in 2D environments like the one highlighted in Fig. 4 c and e, where the agent needs to recombine information from several sequences that occurred during learning and complement this information by inferred knowledge about connections that were never encountered during learning, based on learnt symmetries of the planning environment, or to control a simulated robot so that it can achieve goals that never occurred during learning (see our last demo). We clarify this point on ll 73 – 80 and ll 230 – 244. We are explaining there why previous approaches that are based on sequence learning cannot solve these types of tasks, and point to the completely new ingredient of our method, the learning

of a cognitive map whose geometry makes it suitable for solving such tasks. We also point out there that the CML learns just as well from randomly chosen triples of the current observation, the current action, and the next observation, and does not require any connected action sequences during learning.

Altogether, the planning approach of the CML is fundamentally different from previous approaches based on learning and replaying sequences, both on the conceptual level (i.e., the top level of the Marr hierarchy), on the algorithmic level (the intermediate level of the Marr hierarchy), and on the implementation level. We have therefore added two paragraphs (ll 44 and ll 64) that make the innovation of our work on the two top levels of the Marr hierarchy more explicit.

- Therefore it is difficult to assess how the proposed approach compares to alternative ones.

We have compared our approach with any previously published approach that can solve the same types of tasks. In particular, we have compared the planning performance of the CML with that of the best offline planning methods that are known, such as the Dijkstra algorithm, see ll 317, 345, 378, 384, 394, 422. We could not find any published methods that can handle the generalization challenges that we are discussing in our subsequent demos. To the best of our knowledge, there is very little prior work on general methods for flexible planning, and even less that only requires local synaptic plasticity for learning.

- For example, a paper on predictive learning that is closely related to the work proposed, but not mentioned is here:

<https://www.nature.com/articles/s41467-021-21696-1>

The authors should explain how their approach compares to the work mentioned above and show where it differs and how it improves that previous approach.

(Three comments here are addressed separately below)

I am providing examples of papers that are related to this, but please keep in mind that this is not an exhaustive set, and the authors should perform a thorough literature search to give credit to previous models and highlight the differences/advantages/disadvantages:

- https://doi.org/10.1162/NECO_a_00893
- <https://doi.org/10.1162/neco.2009.11-08-901>
- <https://doi.org/10.1523/JNEUROSCI.4098-12.2013>
- <https://doi.org/10.3389/neuro.10.023.2009>

- <<https://doi.org/10.1371/journal.pcbi.1010233>>

All these papers discuss methods based on learning to predict or to replay sequences. We could not find in any of these papers methods for solving the types of tasks that the CML solves, see the preceding description of them.

- The authors should explain how their approach compares to the work mentioned above and show where it differs and how it improves that previous approach.

As mentioned above, the cited work is not related to our approach, and the methods that have been introduced there cannot solve any of the planning tasks that are considered in our paper. Specifically, with these methods an agent cannot produce close-to-shortest plans for reaching any given goal node from any given start node in a general graph (see our first demo), solve planning challenges in 2D environments like the one highlighted in Fig. 4 c and e, where the agent needs to recombine information from several sequences that occurred during learning and complement this information by inferred knowledge about connections that were never encountered during learning, or to control a simulated robot so that it can achieve goals that never occurred during learning (see our last demo).

- More generally, the claim that there is a lack of compelling models for brain inspired sequence learning and problem solving is not totally correct, as there are several models that in one way or another have addressed (although partially) such challenge.

We have not claimed that there is a lack of compelling models for brain inspired sequence learning. As explained above, none of the cited work on sequence learning can solve the types of problems that we tackle in this manuscript.

- For this aspect, the authors should explain how their work differs, or extends previously published models.

We have added on ll 44 and 64 two new paragraphs in the Introduction which explain the innovation of our approach on the conceptual and algorithmic level. On ll 73 we have added an explanation why this approach is unrelated to the learning or replay of sequences.

- The authors do mention that models of goal-directed action sequences have been indeed proposed in the Reinforcement Learning literature, but they do not explain what the differences are with respect to the approach proposed. It would be useful to have a review of the models that are closest to the presented one and to clarify their advantages/disadvantages.

We are grateful for this suggestion, and we have added such a detailed discussion Reinforcement Learning (RL) on ll 620 - 668 of the Discussion, addressing in particular RL models that are closest to the CML.

The authors should also compare their model with related work on neural network approaches to Constraint Satisfaction Problem solving, including the neural sampling framework of the same group (but, e.g., see also <https://doi.org/10.3389/frai.2020.580607>).

In addition there have been several publications on neuromorphic systems for these types of problem-solving tasks that the authors should also consider:

- <https://doi.org/10.1073/pnas.1212083110>
- <https://www.nature.com/articles/ncomms9941>
- https://doi.org/10.1162/NECO_a_00785

A common aspect of the CML and sampling methods for solving Constraint Satisfaction Problems (CSPs) is that both provide methods for solving specific types of problems. But the class of problems that can be solved by each method are very different. We have added on ll 583 an explanation of the type of problems that a CML can solve and which ones it cannot solve.

Another difference is that once the CML has learnt a problem space, it can solve a large number of related tasks that arise in this problem space. Also, its latency for problem solving tends to be much smaller than the latency of problem solvers based on sampling.

- Based on these comparisons they should revise their claims, that nothing exists yet (already in the abstract) all the way up to the discussion (e.g., lines 528-530).

We would be happy to revise this claim, if we would become aware of relevant prior work. The papers that were cited by the reviewer do not provide a counterexample to our claim because they address different problems, as explained above.

- As the claims are that the model proposed is inspired by real neural circuits, can be used to solve planning tasks in physical environments and be applied to robotics and real-time neuromorphic hardware, the authors should formulate the model and equations in continuous time rather than discrete time.

It would be more adequate to say that the CML is inspired by results from cognitive science about cognitive maps, To the best of our knowledge, the concrete neural circuits of the brain which learn and represent these cognitive maps remain opaque, especially for the human brain.

The recent review

Frenkel, C., Bol, D., & Indiveri, G. (2023). Bottom-Up and Top-Down Approaches for the Design of Neuromorphic Processing Systems: Tradeoffs and Synergies Between Natural and Artificial Intelligence. *Proceedings of the IEEE*.

lists numerous different neuromorphic hardware approaches. But only a minority of them employ continuous time. In particular, SpiiNNaker from the University of Manchester, Tianjie from Tsinghua University, and most neuromorphic hardware approaches of major industrial companies, such as Intel and IBM, employ discrete rather than continuous time (to be precise; intel employs discrete time steps of flexible length, for which our approach is well-suited). Unfortunately it is difficult to implement algorithms that are formulated in continuous time without performance loss in any of these prominent neuromorphic hardware approaches that employ discrete time steps. However, all of these neuromorphic chips that operate in discrete time have been used to control physical robots. Hence, we are confused by the remark of the reviewer, which seems to make the claim that only neuromorphic chips that operate in continuous time can be used to control robots.

On the basis of these facts it does not appear to be meaningful or advisable to make our CML approach substantially more complicated and harder-to-understand by formulating it in continuous time. This would also obscure its conceptual and algorithmic underpinnings, which are the heart of our paper. In particular, this would make it substantially more difficult to compare the CML with other planning methods and relate it to models for cognitive maps in the brain such as

Whittington, J. C., Muller, T. H., Mark, S., Chen, G., Barry, C., Burgess, N., & Behrens, T. E. (2020). The Tolman-Eichenbaum machine: unifying space and relational memory through generalization in the hippocampal formation. *Cell*, 183(5), 1249-1263,

since these other methods and models operate in discrete time.

In fact, also standard literature on planning in the brain, see the review by (Mattar and Lengyel, *Neuron* 2022), is formulated in discrete time.

In order to address this comment of the reviewer we have made in the section “Considerations for the implementation of CMLs in neuromorphic hardware” on pp 21 completely clear for what types of neuromorphic hardware the CML is suitable. Fortunately, this encompasses the neuromorphic hardware that is developed by major companies such as Intel and IBM.

We would like to point out on the side that a CML can operate with a very fine discretization of time and actions, since the selection of the next action according to Principle II can be carried out with very small latency and little computational effort especially when memristor arrays (or “in memory computing”) are employed. Hence running a CML with a fine time discretization is from the functional perspective hard to distinguish from an approach that operates inherently in continuous time.

- This will unveil potential problems related to causality already starting from equations (2) and (3), e.g., when are the various signals at $(t+1)$ actually produced? which one arrives first? what happens between instants (t) and $(t+1)$? how can signals at (t) be multiplied by signals at $(t+1)$? how are these signals kept around? what are the required time-constants of the system? how do they relate to the dynamics of the input signals? what are the delays (if any) needed to support the model? This is a critical point that could potentially invalidate the whole model.

We cannot follow the reviewer in the conjecture that a continuous time formalization could unveil potential problems related to causality. We are not aware of any problem related to causality in the CML approach. But we also agree that this was harder to see without an example for a possible circuit implementation, like the one which is now provided in Fig. S1. In particular, this implementation provides answers to all questions that the reviewer has listed above.

How signals can be delayed or stored for one discrete time step in NMHW depends on details of the hardware. One can for example implement such a delay by employing intermediate relay neurons, in particular inhibitory interneurons. Note that all signals in the implementation of Fig. S1 that are subject to a delay are also subject to a sign-change. This sign-change can be implemented by interneurons that simultaneously produce the unit-delay of the signal.

In addition, NMHW that operates in discrete time usually contains digital registers, and these can also be used to implement a one-step-delay.

- In addition to the critical discrete-continuous time issue, it would be helpful, just for sake of clarity, to have a neural network diagram with the connectivity between the units representing the observation signals, the action signals, etc.

Such a neural network diagram is provided in Fig. S1, and explained in section 1 of the Supplement.

- The authors make a strong point (already from the title) about their model using a local synaptic plasticity rule that is compatible with biology and neuromorphic hardware friendly. But for other parts of the model it is not clear if this locality principle applies. For example the weight normalization of eq. 5 appears to require knowledge of all values of a whole column. The authors should clarify if this corresponds to all inputs afferent to a neuron (e.g., referring to the neural network diagram figure requested above) or to all neuron activations. This would allow them to discuss about biologically plausible mechanisms for achieving such normalization (e.g., homeostatic plasticity, synaptic scaling, etc.), but it would also force the authors to discuss about the time scales and dynamics of this

process (e.g., compared to the dynamics of learning) which are crucial for both biology and hardware implementations.

This is a very good point, and we are grateful to the reviewer for bringing this up. We have redone all experiments, and verified that the weight normalization of eq. 5 can be dropped, provided one works in a sufficiently high-dimensional state space. We have discussed this in section 3 of the Supplement and in Fig. S2. It turns out that weight normalization is not needed if one employs a state space with a dimension of at least 1000 (4000 for the ant demo). This amounts to representing states by 1000 (4000) neurons. Hence operating in these higher dimensional state spaces does not cause a problem for implementations in neuromorphic hardware or brains, where networks consisting of a few thousand neurons or more are standard.

- Another global mechanism is the WTA operation of eq. (7). The authors should explain why/how this does not invalidate the advantages of the local learning mechanism used.

It is now clear from the new Fig. S1 that WTA, which occurs in panel b, is not involved in leaning (which is presented in panels a and c).

With regard to implementing the WTA operation: WTA operations are commonly used in various different neuromorphic hardware designs, In fact, if one enters “neuromorphic winner take all” in Google Scholar, one gets a list of so many publications that we cannot cite them here. One prominent example from INI in Zürich is

R. Kreiser, T. Moraitis, Y. Sandamirskaya, G. Indiveri On-chip unsupervised learning in Winner-Take-All networks of spiking neurons, BioCAS 2017 424-427, 2018.

We also want to point out that it is not essential for the planning performance of the CML that the WTA operation is carried out with high precision: If several actions have high eligibility it has little impact on performance which one of them is chosen. In addition, the WTA operation only needs to be applied to currently affordable actions, hence in general not to a large set of options.

- Similarly, they should further clarify if/how the element-wise multiplication of eq. (6) can be done in parallel, locally without requiring sequential CPU-like processing.

Yes, this element-wise multiplication can be done in parallel where each neuron whose activity represents the estimated utility of a specific action is inhibited by a specific neuron according to how much this action would not be “affordable” in the

current state. If affordance values assume only values 0 and 1, as is the case for most of our demos, it suffices to completely inhibit each neuron that represents the utility of an action with affordance value 0. There also exists a large literature on models for multiplication in neural circuits of the brain that do not require sequential CPU-like processing.

- To their benefit, the authors do mention in-memory computing and cross-bar arrays of memristors that could support this operation in principle,

We have discussed with Dr. Ortner from IBM in Zürich a possible implementation of CMLs in their neuromorphic hardware, that is based on memristor arrays. We have reported the result of this analysis on II 489. According to this analysis the CML can be implemented very efficiently on the IBM hardware for in-memory-computing, with very low latency, and low energy consumption.

- but then they claim that their model could be mapped onto the Loihi neuromorphic platform, which does not use such cross-bars.

We are not aware of making any claim that the CML can be implemented on a Loihi chip with crossbars, to which the reviewer apparently alludes. Rather, we stated that CMLs appear to present no problem for implementation on Loihi or for in-memory computing.

With regard to an implementation on Loihi: We have discussed the implementation of CMLs on Loihi 2 with Dr. Danielle Rager from the neuromorphic team at Intel. According to her analysis, there are no obstacles for implementing CMLs efficiently on Loihi 2. In particular, the type of synaptic plasticity that the CML employs according to equations 2 and 3 can be implemented through on-chip learning on Loihi 2. In fact according to experts from Intel this is at the moment one of very few functionally powerful applications for on-chip learning on Loihi 2.

- To be really convincing that the model can be implemented on energy-efficient neuromorphic hardware the authors should be more precise and substantiate these claims with detailed specifications.

We have provided a neural circuit design for the CML in Fig. S1, and discussed implementation details in the section on pp. 21. But, as we emphasize in the Introduction, the reader first needs to understand the completely new conceptual and algorithmic approach that we present, and which is independent of implementations. In our view it would be best to leave “detailed specifications” to a follow-up paper on an implementation of the CML approach in concrete neuromorphic hardware. One

such paper, where we present the implementation of CMLs on Loihi 2, is already in the works.

.

- On a side note, both the Loihi platform and the SpiNNaker platform mentioned in the manuscript were not designed to minimize energy consumption (e.g, the Loihi chip has 2 standard 8086 CPUs that burn the standard CPU power consumption). So in that respect the authors should not refer to them as "energy efficient" neuromorphic systems.

We cannot follow the reviewer here, because this remark is in contradiction to published literature. Both the designers of Loihi and SpiNNaker state in publications that minimizing energy consumption is an essential goal of their neuromorphic hardware..

The remark of the reviewer is also in contradiction to actual measurements of the energy consumption and EDP product of implementations of various algorithms on Loihi 1, which were published for example in

Davies, M., Wild, A., Orchard, G., Sandamirskaya, Y., Guerra, G. A. F., Joshi, P., ... & Risbud, S. R. (2021). Advancing neuromorphic computing with loihi: A survey on results and outlook. *Proceedings of the IEEE*, 109(5), 911-934.

and

Rao, A., Plank, P., Wild, A., & Maass, W. (2022). A long short-term memory for AI applications in spike-based neuromorphic hardware. *Nature Machine Intelligence*, 4(5), 467-479.

- In addition to comparing to the classical computer science algorithms as benchmark (e.g., Dijkstra's algorithm) the authors should compare performance with other neural network approaches (e.g., see literature mentioned above, but also the e-prop work from the same group).

We examined the literature that the reviewer proposed, and we could not find there any method or result for solving planning problems of the type that are addressed in our manuscript. In particular, we cannot find any competing neural network approach.

The e-prop method is not needed for implementing CMLs since they only require local synaptic plasticity.

- Benchmark comparisons with specific accuracy numbers have become the standard (for better or worse) in recent literature and they should be added here as well.

We have provided comparisons with the performance of the Dijkstra algorithm, the gold standard for planning. We have shown that CMLs are quite competitive with this offline planning method, in spite of the fact that they employ online planning, which has the advantage to produce actions with lower latency and provides flexibility when the goal changes.

- Some choices made by the authors are not properly justified (e.g., why 200 and 32 at lines 222-223?), and some details are missing (e.g., "after a while" on caption of Fig. 2e).

Thank you for your suggestion. We have added our rationale behind choosing these numbers on ll 693 - 694. The caption of Fig. 2e has also been revised to clarify any ambiguities.

- The caption of Fig.2f does not explain what the figure is about. It is more a comment or an observation on the data. It is not clear from Fig.2f if the error goes all the way to zero and would stay at zero if learning were to continue (i.e., if the network stays in the global minimum once it finds it or if it can jump out of it later).

We have added a sentence to the caption of Fig 2f which clarifies what it shows. Also ll 298 of the text explain that. Unfortunately, the reviewer has misunderstood what it shows. It is not related to the energy function of a CSP, or to any other global minimum of any sort.

- Section at line 263 should relate the problem presented to the "Traveling Salesman Problem", and at the end of that part it would be interesting to see if the model proposed could be used to solve generic SAT problems, to compare with standard benchmarks used in the CSP literature.

The CML is not a generic CSP solver. It is made to solve planning problems and other problems as discussed in chapter 3 and 11 of (Russell and Norvig, 2020) and in the review of (Matta and Lengyel, 2022) of related models from neuroscience. These planning tasks are special cases of optimization problems that can be formulated as shortest path problems in general graphs. To the best of our knowledge the Traveling Salesman Problem cannot be formulated in this way. Furthermore, the planning problems that we consider fall into the complexity class P, whereas the Traveling Salesman Problem is NP complete, and hence requires other methods.

- The section starting at line 389 should clarify that the robotic demonstration is *virtual* and no real robot was used in this work. The authors should make it clear from the title that it is a *simulated* *virtual* experiment.

Done (see II 110, 454 and 456)

- The section on Neuromorphic Hardware should be substantiated with more details (see comments above), for example explaining how to map the model (how many neurons, how many synapses per neuron, how many connections, how to implement the WTA, etc.).

We have provided in section 1 of the Supplement and Fig. S1 a detailed account of a possible implementation of the CML with neural circuits. The requested numbers are provided there in Table S1.

- Typical neuromorphic hardware platforms (and even ANN accelerators) try to reduce the precision of the weights to 4 or 3 bits. So the requirement to use more than 6 bits severely restricts the possibility to implement such model on generic neuromorphic hardware and limits it to only the current generations of SpiNNaKer and Loihi (as future optimized ones will likely reduce the weight bit resolution).

We cannot reconcile this remark of the reviewer with recent results of the Yang Lab that have been published in Nature:

Rao, M., Tang, H., Wu, J., Song, W., Zhang, M., Yin, W., ... & Yang, J. J. (2023). Thousands of conductance levels in memristors integrated on CMOS. *Nature*, 615(7954), 823-829.

and with methods for bit-slicing for memristor technologies that provide lower bit resolution, see II 523. The latter makes it for example possible to implement CMLs with 8-bit precision of weights on the Hermes Chip from IBM.

- The section on the relation to transformers is tangential to the work proposed and (in an effort to reduce size) could be moved to a supplementary material part.

We beg to differ: We point out in this subsection that the CML can be viewed as a low-dimensional projection of Transformers. This link to Transformers will be of interest to a large part of the AI and NMHW community readers that is particularly interested in lightweight variants of Transformers that are suitable for efficient implementations in NMHW. In fact, we find the structural similarity of fundamental equations of Transformers and CMLs that are elucidated in this section quite striking. Hence, they are likely to inspire further work on light-weight Transformers versions and their implementation in NMHW.

- The discussion should highlight even more (perhaps bringing that argument further up) how the model proposed requires "only" Hebbian plasticity and does not need back-propagation.

We are grateful for this suggestion, and have mentioned this feature already in ll 9 of the Abstract, and also in ll 66, 86 of the Introduction.

- There is not sufficient details to understand how to reproduce the results. The authors should provide a more comprehensive description in the methods and provide access to the code.

We are making together with this revised version of the ms our code for applications of CMLs in general graphs public. We have also made efforts to make sure that our description of algorithms are transparent and complete. These efforts include:

- On ll 160-162 and in Equ. 4 we modified the formulation of Principle II so that it now becomes clear that $u_t(a)$ is a scalar function, and its value changes for different actions a .
- On ll 677-679, we addressed possible confusions regarding Equation 11.
- On ll 699-709, we unified the parameters used on all abstract graph tasks, and detailed the rationale behind their selection. We did the same for the 2D navigation tasks on ll 746-751, and on ll 766-772 for the quadruped controlling task.
- On ll 710-718. we listed all parameters one needs to tune the CML algorithm, and discuss their robustness.
- We redesigned Figure 1, so that the computational graph becomes more clear.
- In Section 1 of the Supplement, in particular in the new Fig. S1, we described a neural network implementation of the CML algorithm, which should aid readers in understanding our approach.

Reviewer #2 (Remarks to the Author):

Review of Stöckl and Maass (2023) "Local prediction-learning in high-dimensional spaces enables neural networks to plan"

- The authors propose a neuromorphic algorithm

Actually, we propose a new conceptual and algorithmic approach for online planning and problem solving, that is based on self-supervised learning through local synaptic plasticity. This innovation addresses primarily the two top levels of the Marr hierarchy. Hence, we would like to ask the reviewer not to reduce our innovation to the implementation level, the 3rd level of the Marr hierarchy. There are obviously many different ways to implement such a new conceptual and algorithmic approach. In order to avoid this misunderstanding, we are now addressing more explicitly the levels of the Marr hierarchy and our contribution to its two top levels in the Introduction (ll 44 and ll 64).

However, we are also providing in the revised version of the paper in the first section of the Supplement and Fig. S1 the design of a possible neuromorphic implementation of a CML.

by which an agent can learn the structure of an environment and decide on the best course of action to reach a given target location in the shortest time. The principle is to embed both the states (graph nodes) and the actions (graph edges) in the same high-dimensional space, such that the graph distances are represented by geometric distances between nodes, and the actions correspond to directions in the space. Then the agent can pursue the shortest route by choosing the action that points in the direction of the target node. The embeddings of both states and actions are learned during random exploration of the graph, based on optimizing a prediction of the next state from the previous state and action. Ultimately the embedding should be such that the next state can be predicted by simply adding the action vector to the previous state. The authors argue that all this can be accomplished with biologically plausible learning rules.

Evaluation:

- This is an interesting suggestion for how an agent might learn the structure of its environment, namely how states and actions are connected. This ultimately results in a goal signal, by which the correct action can be chosen that leads to the goal in the shortest amount of time. Some of my questions concern whether the algorithms are really neuromorphic, in the sense that they could be implemented in biological neural circuits.

We have added in Fig. S1 a possible way to implement our new method in a neural circuit. We have also added two new sections to the Supplement, entitled “Possible implementation of CMLs by neural networks” and “Linking neuronal circuit implementations of CMLs to experimental data from neuroscience”, which discuss this neural circuit implementation and its relation to biological data.

Concerns about bio-plausibility:

- 1. The learning rule for state-action association, Eqns 2 and 3: This looks difficult to implement with neurons, because:
 - a) it involves two different time points on the post-synaptic side: s_{t+1} appears at time $t+1$, but \hat{s}_{t+1} at time t ;

The timing details of the proposed neural circuit and its learning rule are specified in the new Fig. S1, in particular in panel a. The error term in the learning rule involves an action command from the preceding time step. Numerous experimental studies show that information from the recent past is provided by the neural activity of many cortical areas. For example, it was shown in

Wang, Z. A., Chen, S., Liu, Y., Liu, L. D., Svoboda, K., Li, N., & Druckmann, S. (2023). Not everything, not everywhere, not all at once: a study of brain-wide encoding of movement. *bioRxiv*, 2023-06

that neural activity in V1 represents information about motor activity in the recent past (30 – 60ms ago). We have added a remark on that in I 132 - 144 of the Supplement.

Altogether, the term to which the reviewer refers, the prediction error, is closely related to forms of prediction errors that have been shown in numerous studies to be represented by the activity of particular populations of neurons. We discuss these links on I 98 – 110 of the Supplement.

- b) to make s_{t+1} appear in that neural population, the action vector must temporarily be switched off, requiring some kind of control system;

We show in the new Fig. S1 a and c that it suffices to delay signals that go through inhibitory relays, no switching off is needed.

- c) the postsynaptic variable is a difference between two successive time points. It is not obvious how to implement such a delta rule with biophysics.

We have added on II 96 - 111 of the Supplement links to biological data which point to the presence of error signals in the brain. On II 82 - 95 of the Supplement we have added links to biological data on synaptic plasticity rules that can implement such a delta rule. A similar delta learning rule is used in Algorithm 2 of (Zhang et al., eLife 2023), and is described there as being biologically realistic. We refer to this recent publication on II 628 because it has somewhat related goals.

We have added references to experimental data on related rules for synaptic plasticity on I 82 - 95 of the Supplement. In particular, we point there to in-vivo data on synaptic plasticity that support plasticity rules of the type that we are using, but not STDP or Hebb since they depend on gating signals rather than postsynaptic activity.

- 2. The computation of utility, Eqn 4: Again, how would this be implemented with neurons?
 - a) It involves multiplying a population vector with another vector in the same population; what would be the mechanism for such a multiply-and-add operation?

In the proposed neural circuit implementation of the computation of utilities in Fig. S1b each of these multiplications is carried by synapses that multiplies their synaptic inputs with their synaptic weights. Note that the elements of V and Q are learnt synaptic weights. The multiply-and-add operation of equ. 4 is implemented there through sequential processing by two layers of synapses in a 2-layer feedforward neural network, one with synaptic weights Q and one with synaptic weights V . The add-operation is implemented through the weighted sum that is computed by each neuron.

- b) In the left term, the activity is driven by an action; in the right term, activity of the same neurons is driven by observations. How would that be accomplished?

A circuit diagram for the computation of Eqn. 4 is shown in Fig. S1b. This computation could for example be carried out in a motor area, where each neuron is linked to a particular action, but receives through its synapses information about sensory inputs. For example,

Zagha, E., Ge, X., & McCormick, D. A. (2015). Competing neural ensembles in motor cortex gate goal-directed motor output. *Neuron*, 88(3), 565-577.

suggests that neurons in the motor cortex form a competitive circuit that regulates sensory-to-motor transformation, similarly as proposed in our Fig. S1b. We have added a remark on that in I 69 – 75 of the Supplement.

- c) The right term involves subtracting activity from the current observations from activity driven by some remembered observation. How are the remembered observations stored? Are these fed into the network one after another? How are they subtracted?

The term in question in eq. 4 represents a difference between a target observation (goal) and the current observation. With regard to target observations or goals, it had been shown in

Basu, R., Gebauer, R., Herfurth, T., Kolb, S., Golipour, Z., Tchumatchenko, T., & Ito, H. T. (2021). The orbitofrontal cortex maps future navigational goals. *Nature*, 599(7885), 449-452.

that future navigational goals are represented by neural activity in the OFC of the rodent (we refer to that result on Il 61 of the Supplement). Differences are commonly computed with the help of inhibitory interneurons in neuromorphic circuits.

- 3. Normalization of synaptic weights

- a) Eqn 5: Here each input synapse gets modified by a factor that depends on the strengths of all the other synapses onto that same neuron. This violates the locality of synaptic plasticity: A synapse should be modified based only on its own strength and the activity of the pre-and postsynaptic neurons.

We are grateful to the reviewer for this remark. It led us to re-examine the need for a weight normalization operation. We found that this weight normalization is not necessary, provided that the dimension of the state space (i.e., the number of neurons that are involved in presenting a state) is sufficiently large, for example between 1000 and 4000 for the tasks that we considered. We have addressed the impact of the dimension of the state space on implicit weight normalization in Section 2 of the Supplement (and in Fig. S2).

- b) Another related normalization appears in line 660.

We found that this normalization is also not necessary if the dimension of the state space is sufficiently large. Therefore we have also eliminated this type of normalization from all our experiments, and found that this hardly affects task performance, provided that the dimension of the state space is not too small (between 1000 and 4000 for the tasks that we considered).

- 4. Winner-take-all choice (Eqn 7): Explain how this would work in a neuromorphic system? How does the agent sample the values of the affordable actions? By actually executing them in the real world one at a time and comparing the resulting output from Eqn 4? But then the observations o_t would change as all the actions are played out. Or is the comparison done “mentally” without real-world action? If so, does this require some accessory system that can look up the codes of all the affordable actions and feed them into the network one at a time?

We have added a sketch of a neural circuit that implements the action selection strategy of the CML in Fig. S1 b. Utilities and eligibilities for all possible actions are computed in parallel, and a WTA circuit selects the action with the largest eligibility. A mental action selection process is consistent with the experimental data of

Zagha, E., Ge, X., & McCormick, D. A. (2015). Competing neural ensembles in motor cortex gate goal-directed motor output. *Neuron*, 88(3), 565-577.

We have added pointers to these experimental data and related models on ll 68 of the Supplement.

- In summary, if the authors advocate that this model could be implemented in brains it would be helpful if the proposal included a bio-plausible neural circuit for each of these operations.

We are grateful for this suggestion, and we have amended the manuscript accordingly. In particular, we have related in section 2 of the Supplement the proposed neural circuit implementation and learning rules to experimental data from neuroscience.

Other questions:

- 5. Graphs with cycles
 - a) As pointed out (p.9), the theory behind this learning model causes problems with cyclic graphs. All the graph edges are supposed to be orthogonal in the embedding space, yet adding the edges around a loop should give zero. Both can't be true.

We had addressed this issue in ll 246 of the submission (now further elaborated it in ll 298 - 309).

- It is not clear how this conflict was resolved. Does it require careful parameter tuning so as to sustain performance of the model?

No, it emerged automatically, and within the relatively short training time that we used. Our understanding is that in a sufficiently high-dimensional state space it is less difficult to satisfy both of these constraints in an approximate way, as Fig. 2 c and f show.

- b) Line 585 describes a hack by which the agent was prevented from traveling cycles during exploration. That requires oracular knowledge of the graph. How would that work in a real-world system?

It suffices to implement some form of inhibition of return. This mechanism has been frequently documented on the level of behavior. On the level of neural circuits, both neuronal adaptation and synaptic depression are mechanisms which are able to implement it.

- 6. The concept of “action”
 - a) In the first part of the paper (p.1-12), every edge on the graph is considered a different action, and they are all encoded with one-hot vectors. For example, a robot action like “turn left” would have a different one-hot code at every different location.

Our 1st demo addresses the standard formulation of general planning and problem solving as path finding in an abstract graph, see for example ch. 3 of the standard AI textbook (Russell and Norvig, 2020). In this literature one refers to the edges of the abstract graph as “actions”, although they refer in general not to movements of a human or a robot. These actions may represent operations that are engaged for reaching more abstract goals, such as “become admitted at country club X”, or “improve the grade of my son in math”. For movements of a robot, such as “turn left”, a state-independent code for an action is more adequate. Therefore we have employed such a type of state-independent action codes in our 2nd demo (navigation) and in our 3rd demo (ant locomotion),

- By postulating a one-hot code for every edge on the graph one effectively circumvents the problem of path integration, which requires making a prediction from stringing together actions that may appear identical from the agent’s perspective.

We have shown that the CML can plan very well, both with state-dependent one-hot codes for actions (see the abstract problem solving in demo 1), and with state-independent one-hot codes for actions (in our demos 2 and 3). In particular, our results for demo 2 and 3 show that the CML is able to plan very well in scenarios where trajectories to the goal involve “actions that may appear identical from the agent’s perspective”. This becomes possible through the learnt geometry of a cognitive map. Hence, we do not see that we “circumvent” a problem.

- b) Please explain how a naive agent entering a new environment will already have a one-hot code for every edge on the graph, even if they involve identical movements of the agent.

The reviewer refers here apparently to the application of the CML in demos 2 and 3, since actions represent movements only in these two demos. As we explained above, we do NOT assume in demo 2 or 3 that there is a separate one-hot code for every edge of the graph. Rather, actions are there state-invariant. We have added on ll 403 - 405 a remark which clarifies this.

We have added before that on ll 397 a comment which clarifies that one can employ CMLs both with state-dependent and with state-independent internal codes for actions.

- c) What if the actions instead were agent-centered, e.g. {left, right, forward, back} as in the second part of the paper (p.13ff). Would the system still be able to learn an arbitrary graph?

This question assumes apparently that edges that leave different nodes in an arbitrary graph are agent-centered, i.e., the same actions can be applied in different nodes. But since outgoing edges of nodes are not labeled in an arbitrary graph in a way that is independent of the node from which they come, we do not see a way to emulate in an arbitrary graph the concept of having agent-centered, i.e., node-independent, actions.

We had addressed the case of agent-centered actions both in demo 2 and demo 3, and we have shown that the CML is able to learn in both cases the given graph. There we challenged the CML to learn given graphs that represent planning in different physical environments. In particular, for spatial navigation we have shown in Fig. 4 and S4 that the CML learns not only graphs that result from movements left, right, forward, back, but also graphs that arise in a different scenario where one has 6 movement primitives that only make sense in a hexagonal environment. The CML learnt cognitive maps both for the case of 4 and for the case of 6 actions. without getting any outside hints about the meaning of these actions. Furthermore, the CML was able to exploit in both scenarios their inherent symmetries for efficient planning. In addition, in demo 3 the CML learnt a cognitive map for moving in a 3D world, a third scenario with inherent symmetries.

- 7. “Affordances”
 - a) What is the source of the “affordances” in Equ 6. They represent part of the graph structure. Don’t these need to be learned as well? In the present formulation it seems they are offered to the system without learning (contra line 194).

We had stated on ll 194 of the old version that no prior knowledge is required for the CML in the case of abstract graphs. We have now made on ll 251 explicit that this remark applies only to the case of abstract graphs, and stated on ll 405 that prior information (given via the affordance module) is needed in physical environments when not all actions can be executed in each state.

Since this paper is already quite full, we found it ill-advised to address here also the learning of affordances, and its complementation through innate knowledge. To the best of our knowledge, we are actually missing clear guidance from experimental data about learning of affordances. Reviews such as

de Wit, M. M., de Vries, S., van der Kamp, J., & Withagen, R. (2017). Affordances and neuroscience: Steps towards a successful marriage. *Neuroscience & Biobehavioral Reviews*, 80, 622-629.

show that the psychological and behavioral role of affordances is well documented, but that there is a lack of precise knowledge how affordances are learnt and represented in neural circuits of the brain. Many brain areas, in particular PFC, appear to be involved. PFC is known to inhibit certain actions in certain states, suggesting that inhibition could be a general mechanism for moving from utilities u_t to eligibilities e_t in the circuit sketch of Fig. S1b.

We agree with the reviewer that affordances are in many cases subject to learning. But apparently there also exists some innate inhibitory control of action selection, since trying out each action in each state tends to be fatal for an organism.

- b) By what mechanism would this happen in a neuromorphic system, and how would they be represented by the neural circuit?

Affordances are represented in eq. 5 by gating factors g for each possible action. These are multiplied with the utility estimate for each possible action. One could represent them through the activation of inhibitory neurons that reduce the activation of neurons that represent particular actions. PV-cells are suitable for binary-valued affordances, and SST-cells are suitable for graded inhibition that encodes graded affordance values. Non-binary affordance values appear in our paper only in 2 tasks: Finding shortest paths in weighted graphs and locomotion of the ant, see the comments on ll 729 and ll 816.

Apart from this implementation option, there exists a very large literature in computational neuroscience that report mechanisms for the implementation of multiplication in neuromorphic circuits, see e.g.

Groschner, L. N., Malis, J. G., Zuidinga, B., & Borst, A. (2022). A biophysical account of multiplication by a single neuron. *Nature*, 603(7899), 119-123.

for a recent report.

c) Figure 4e highlights the fact that the agent takes an action that was never taken during exploration (Fig 4c). But what if there was a wall between those two nodes, and that's why the action never occurred there? How does the agent know that the action is "affordable".

We assume that such information is provided by an affordance module that is not part of our model. We are stating this explicitly on ll 226 and ll 743 of the main text, and ll 118 of the Supplement. In general, an adaptive module that learns affordances could easily be added to the CML model, using for example visual input or whiskering information in order to detect a wall. But this would obviously be beyond the scope of this paper.

- 8. Parameter sensitivity
 - a) It appears that for each of the illustrated graphs the embedding space had a

different dimensionality, and different choices were made for the learning rates (p.22). In line 610, the state space has fewer dimensions than the number of actions, which obviously precludes finding orthogonal vectors. What motivated all these different choices?

We agree that this was a defect of the first version, where we had not made any effort to use uniform parameter values. We now give on ll 710-718 an overview of all parameters that are relevant. We report on ll 699-709 values of parameters that worked for all abstract graph tasks, and also for all 2D navigation tasks (see ll 746-751 for the latter). Only the ant control task requires somewhat different parameters since it has to deal with continuous-valued observations and a very large set of state-independent actions, see our remarks on ll 766-772. However, the same dimension 4000 of the state space can be used for all tasks that we discuss in the paper.

- b) How robust is the system to the relevant parameters? Can the same agent with one parameter set learn different spaces? The report would benefit from exploring a wider range of graphs and scanning over parameter values.

Yes, it is possible for an agent to learn different tasks (spaces) with one parameter set (added on ll 767), but this may not be optimal for some tasks are apparently easier and needs fewer dimension of the state space. We have added in Fig. 2 g and Fig. S2 a scanning over the dimension of the state space on the random graph task. We also report ranges of parameter values that we found to work well on ll 699 and 708 for the abstract graph tasks, on ll 749 for the 2D navigation tasks.

- 9. Related work

There is prior research on these topics that could be used to put the current work in context. In particular, various versions of model-based reinforcement learning acquire the structure of the state space through learning, so that a goal-directed policy can be superposed on that. The present paper only deals with learning the state space, not with learning the goal locations: Those are provided by some unspecified accessory system.

We agree with the comments of the reviewer, and we have strongly expanded our discussion of related RL methods on ll 620. We also agree that learning of goal locations is an important topic, and we have added on ll 634 a reference to recent work on that.

- Within the RL literature there has been recent enthusiasm about the "successor representation", which is an embedding that helps predict the agent's next state. For neuromorphic models that learn the successor representation, see for

example Fang 2023 (<https://doi.org/10.7554/eLife.80680>) and literature cited there.

We are grateful for this reference, and we are addressing relations to this approach on II 643

Other suggestions:

- 10. There is frequent reference to “planning”. But the model presented here doesn’t make any plans.

We have based our terminology on standard literature: The chapters on planning (Chapters 3 and 11) in the standard textbook for AI by Russell and Norvig, and on the apparently best review of results and models for planning in the brain that is currently available: the review by Mattar and Lengyel in Neuron 2022.

According to the section on “online planning” that starts on p. 389 of the standard AI textbook (Russel and Norvig, 2020), the CML is a special case of an online planner.

The review by Mattar and Lengyel defines on p. 923:

“In “online” (also known as “decision time”) algorithms, planning and acting are interleaved, such that the results of planning are used only for choosing the immediate next action...”.

Obviously the CML carries out online planning according to this definition. Although it only produces one action at a time, it chooses this action with “thinking ahead”, since it chooses this action on the basis of its angle with the **direction of the goal** in the learnt cognitive map. In previously considered planning approaches this type of look-ahead was not available, except for planning navigation in 2D.

We hope that the reviewer agrees with us that it is desirable to employ a terminology that is consistent with standard textbooks and reviews.

- Once it arrives at a node it decides on the next action, as though it had a lookup table (see line 13). It does not “think ahead”. In the neuroscience and robotics field, “planning” usually concerns evaluating the outcome of successive actions ahead of time, for example comparing the value of different routes.

We do not find ground for a statement that the CML does not “think ahead”, since it chooses each action on the basis of its angle with the direction of the goal in the learnt cognitive map. Hence, by choosing an action where this angle is minimal it certainly applies “foresight” and “look-ahead”.

One should also take into account that the CML produces in this online manner trajectories to the goal whose length is **close to that of the optimal solutions that are produced by the best offline planning methods that explicitly “think ahead” and try out all possible routes**, see II 316 and II 344. This fact suggests that the geometric form of “thinking ahead” which the CML employs suffices for such tasks.

- The behavior implemented here is more like “online navigation”. This may be confusing to the reader.

The term “navigation” is in our view not adequate, and generally not used, for describing general planning and problem solving challenges, such as those formalized through abstract graphs (see the corresponding chapters in the standard AI textbook by Russel and Norvig). This appears to be justified by the fact that in general actions are in this abstract setting not related to movements in a physical space.

Apparently the term “navigation” is also not commonly used for control tasks where the agent controls the joints of a simulated legged robot, as in our demo 3.

- 11. In Figures 4 and 5, is there a meaning to the emojis painted on the nodes? At least this reader finds them distracting. Preceding figures worked just fine without emojis.

One needs here labels for different observations which clarify that these are pairwise different. Labeling different observations by numbers 1,2, 3, ... or letters a, b, c,... could incorrectly suggest that they have an implicit or explicit order. Therefore we have adopted a labeling of different observations that is commonly used in other work on cognitive maps, such as Fig. 1 of

Whittington, J. C., Muller, T. H., Mark, S., Chen, G., Barry, C., Burgess, N., & Behrens, T. E. (2020). The Tolman-Eichenbaum machine: unifying space and relational memory through generalization in the hippocampal formation. Cell

- 12. In part 2 (p.13ff) the main difference from part 1 is that the actions are now agent-centric. It may help the reader to point this out. Because there are only 4 action vectors, the predictions are forced into a 4-dimensional subspace of the high-dimensional space. Eventually, the learning process squeezes that into a 2-dimensional subspace that accurately reproduces the geometry of the graph.

Done (II 405)

- 13. Eqn 4: should start with $u_t =$

Done.

- 14. Figure 6c: Perhaps show a bit more of the time course to document that the two variables have settled.

Done

- 15. Fig 2 caption: “square root of the length of the shortest path” should this be “square root of the sum of squares along the shortest path”. As stated, it doesn’t resemble the law of Pythagoras.

Done.

- 16. Fig 4a: use different arrows for the 4 actions.

Done.

- 17. Typos:

Line 68 fixed

Line 32 fixed

Line 250: Meaning of this sentence unclear. fixed

Line 392 fixed

Line 505 fixed

Eqn 12: should be V_a instead of V ? explained

REVIEWERS' COMMENTS

Reviewer #1 (Remarks to the Author):

The authors have clarified many of the issues raised, also adding to the manuscript useful discussions and clarifications. However there are still some issues that remain open.

(1) Previous relevant literature: the authors claim that "could not find any methods in the literature that could solve the same tasks as the CML." This is indeed correct, as the authors chose to use very specific examples (e.g., in Fig. 4) that do not appear in previous works. However, they fail to recognize that some of the papers that I pointed out in the previous round of reviews (e.g., the one I explicitly said is closely related to the work presented here - [Recanatesi et al, 2021] <https://www.nature.com/articles/s41467-021-21696-1> -) are indeed very similar and do not only "discuss methods based on learning to predict or to replay sequences.") Indeed, in the paper mentioned, the authors use different language and terminology, but refer to very similar "cognitive map" representations that include both states and actions, and to very similar general problem solving abilities, such as solving a card-game task, which is clearly not a simple sequence recognition or generation task. In addition, despite the different languages used, there are very strong similarities from the point of view of behaviors of both models. For example in the [Recanatesi et al.] paper the authors write that their model "solves this predictive task when, prompted with a pair (ot, at), it correctly predicts the upcoming observation ot+1." which is strikingly similar to the "Principle 1" of the proposed work. Even the neural network diagram of the new figure S1 of this work is strikingly similar to the network depicted in Fig. 1c of the [Recanatesi et al.] paper. So I do not agree with the claim that there is no prior similar work, because they only discuss methods on "learning to predict or to replay sequences". As the authors failed to see the similarities with the [Recanatesi et al.] work, I know wonder if they failed to find similarities with other previous publications that might provide similar methods, but in different context or with different language (e.g., with probabilistic graphical models or time-continuous RNNs).

The claim made by the authors that "The papers that were cited by the reviewer do not provide a counterexample to our claim because they address different problems, as explained above." is not a justification, because the principles of learning to plan can be demonstrated with many different problems, and simply picking specific examples that have not been published before is not a reason to dismiss previous literature.

(2) Continuous time vs discrete time: as the authors argue that the proposing a model that could explain how animal brains solves planning tasks, then the continuous time aspect and temporal dynamics play an important role. I agree with the authors that a simplified discrete-time formulation is sufficient to demonstrate the main features of the CML approach, but a discussion on what extra conditions are necessary to demonstrate that the model can work in real (analog) physical systems such as animal brains would add value to the paper. For example, what would be the constraints on the delays in order to get both signals representing "at" and "ot" available at the same time in the summation module? How much difference can be tolerated? Or, how would the real biological system know when action "A" is

finished and a new action "A" starts (assuming continuous time, with no discrete jumps from "t" to "t+1")?

(3) Neuromorphic hardware: The clarification that the model could be implemented in digital neuromorphic processors is very useful. However I believe that also the IBM chips that use memristor arrays operate in discrete/clocked time. If this is the case the authors should specify this, because otherwise using the term analog might lead the reader to believe that the implementation of the model would work in continuous time.

(4) Energy efficiency: the authors should be very careful with claims on low power. For example, if you read carefully the Loihi paper mentioned, you will find claims such as "Neuromorphic computing *aims* to [...] at low power levels." in the abstract. But if you search for concrete numbers, you will find figures of >10 Watts for *dynamic* power (i.e., removing all the substantial power consumption figures of all the circuits that burn power even when the network is not running). Even so, this *dynamic* figure is still much lower power than GPUs and cloud servers, but it is much higher power than a dedicated digital ANN accelerator for deep networks. The latest ANN accelerators can demonstrate high accuracy classification with less than 1W and sometimes even in the mW range. So even if *in principle* neuromorphic HW could be designed to be very low power, the SpiNNaker and Loihi platforms should not be listed as "low power" chips. To their defense, they were not designed to minimize power consumption. Indeed, a full SpiNNaker system with 1200 boards should consume about 90 kW.

(4) 8-bit vs 4-bit precision: as for claims on energy efficiency the authors should be careful when citing papers and figures on equivalent bit-resolution of memristive devices. It is well known that memristive devices have high variability and cannot achieve 8-bit resolution during learning. The paper that the authors cite from the group of Joshua Yang is achieving a remarkable result, but only thanks to a "denoising" process applied only for inference when the true desired weight matrix is known. This is an iterative process involving multiple reads and multiple writes that would not work in an on-line learning setup. Also other techniques (e.g. with the Hermes chip) will require extra operations and overhead to achieve 8-bit resolution figures. So the authors should be more cautious in making claims about 8-bit implementations in hardware that has memristive devices. The model proposed is already very strong on its own grounds. There is no need to make bold statements that might not be accurate if one were to actually try to implement it in hardware.

(5) Transformers comparison: I read the authors arguments and agree with them on this point.

In summary, overall the paper has improved significantly. Despite the long review discussion, I believe there are very few minor edits to add to the already very strong and convincing manuscript, to address the issues raised above.

Reviewer #2 (Remarks to the Author):

The revised manuscript answers most of my initial concerns. A few remaining suggestions:

Line 165, “they do not depend on any specific policy or goal”: But Eqn 4 explicitly depends on the goal o^* .

Line 219, “affordance”: First appearance of the term in the text. Needs a definition in the present context.

Line 228, “affordance values arise in the brain from a mixture nature”: Meaning?

Figs 4 and 5: I had flagged the use of emojis in Figs 4 and 5 and the supplement as a distraction. In response the authors write (line 409), “we did not label different observations by numbers or letters because this might suggest a linear order of the observations, which does not exist”. This is hard to understand because

a) Just about every paper about graphs indexes the nodes with integers, without implying any order.

b) When labeling the actions A, B, C, D, the authors don't seem to have the same concern.

c) The emojis have exactly the opposite effect: the reader is immediately led to inspect the pictures for any meaning, in particular whether two pictures are the same.

d) In the rebuttal the authors refer to Whittington 2020, a paper that makes copious use of emojis. My students just nominated that paper in a “5 worst figures” contest, so I don't consider it a good role model.

Line 413: “Note that this encoding of actions is actionable”. Meaning? Please explain, or use different words.

Fig 4a: My suggestion “use different arrows for the 4 actions” meant the arrows should point right, up, left, down. Not different hash marks.

Typos:

Line 425, “tested in on”

Line 950

Supplement lines 26-27

Rebuttal to the reviewer comments on our revised ms

We would like to thank the reviewers for their helpful criticism, which has led to further improvements of the submission.

REVIEWERS' COMMENTS

Reviewer #1 (Remarks to the Author):

The authors have clarified many of the issues raised, also adding to the manuscript useful discussions and clarifications. However there are still some issues that remain open.

(1) Previous relevant literature: the authors claim that "could not find any methods in the literature that could solve the same tasks as the CML."

This sentence in the rebuttal referred to our literature search for related approaches, where we meant in this context online planning capability based on prediction learning. Our statements in the ms itself are more carefully worded. Also, the title of our paper makes clear what this paper is about.

This is indeed correct, as the authors chose to use very specific examples (e.g., in Fig. 4) that do not appear in previous works.

This criticism is unfounded. The planning tasks that we considered in Figures 2 and 3 (general graphs) and in Figures 4 and 5 (navigation in physical environments) are standard benchmark tasks for planning, not "very specific examples". The point is that the papers which the reviewer cited do not solve ANY planning tasks.

However, they fail to recognize that some of the papers that I pointed out in the previous round of reviews (e.g., the one I explicitly said is closely related to the work presented here - [Recanatesi et al, 2021] <https://www.nature.com/articles/s41467-021-21696-1> -) are indeed very similar and do not only "discuss methods based on learning to predict or to replay sequences.") Indeed, in the paper mentioned, the authors use different language and terminology, but refer to very similar "cognitive map" representations that include both states and actions, and to very similar general problem solving abilities, such as solving a card-game task, which is clearly not a simple sequence recognition or generation task.

[Recanatesi et al, 2021] claims to "solve a card game task". But they actually just predict the next game state based on the current game state and the current action. This is not a planning task, it is a standard prediction task.

In addition, despite the different languages used, there are very strong similarities from the point of view of behaviors of both models. For example in the [Recanatesi et al.] paper the authors write that their model "solves this predictive task when, prompted with a pair (ot, at), it correctly predicts the upcoming observation ot+1." which is strikingly similar to the "Principle 1" of the proposed work. Even the neural network diagram of the new figure S1 of this work is strikingly similar to the network depicted in Fig. 1c of the [Recanatesi et al.] paper. So I do not agree with the claim that there is no prior similar work, because they only discuss methods on "learning to predict or to replay sequences".

We beg to differ. They only show that their NN learns to predict the next state of the game, based on the current state and action. This is not a planning task. It is correct that [Recanatesi et al.] engages a predictive learning principle similar to the one that we are using in Principle I. There are in fact hundreds of publications on learning to predict. We do not claim any novelty for that, and therefore did not cite literature on learning to predict, except for 3 representative examples on ll 598. This is clearly explained on ll 597:

"Learning to predict future observations, which is the heart of the CML, has already frequently been proposed as a key principle of learning in the brain \citep{rao1999predictive, friston2018does, keller2018predictive}. But it had not yet been noticed that it also enables problem solving and flexible planning."

We also would like to refer again to the title of our paper.

As the authors failed to see the similarities with the [Recanatesi et al.] work, I know wonder if they failed to find similarities with other previous publications that might provide similar methods, but in different context or with different language (e.g., with probabilistic graphical models or time-continuous RNNs).

We are not aware of work in these different languages that addresses the topic of this ms, which is clearly stated in its title.

The claim made by the authors that "The papers that were cited by the reviewer do not provide a counterexample to our claim because they address different problems, as explained above." is not a justification, because the principles of learning to plan can be demonstrated with many different problems, and simply picking specific examples that have not been published before is not a reason to dismiss previous literature.

This criticism is unfounded, as we have already explained above. It is not true that we "simply picked specific examples to dismiss previous literature". Rather, we showed how standard benchmark planning problems can be solved with our approach. In contrast, the papers that the reviewer cited do not solve ANY

planning tasks. In fact, as far as we could see, they do not even claim to solve any planning task.

(2) Continuous time vs discrete time: as the authors argue that the proposing a model that could explain how animal brains solves planning tasks, then the continuous time aspect and temporal dynamics play an important role. I agree with the authors that a simplified discrete-time formulation is sufficient to demonstrate the main features of the CML approach, but a discussion on what extra conditions are necessary to demonstrate that the model can work in real (analog) physical systems such as animal brains would add value to the paper.

We are grateful for this suggestion, and we have added on II 144 of the Supplementary Information a reference to the commonly assumed hierarchical organization of motor control in mammals, as discussed for example in

Arber, S., & Costa, R. M. (2018). Connecting neuronal circuits for movement. *Science*, 360(6396), 1403-1404.

Action selection in our abstract model corresponds best to the top level of this hierarchy, i.e., to the motor cortex. Details of motor control in continuous time are apparently left to lower levels of hierarchical motor control, such as the midbrain and the spinal cord.

For example, what would be the constraints on the delays in order to get both signals representing "at" and "ot" available at the same time in the summation module? How much difference can be tolerated? Or, how would the real biological system know when action "A" is finished and a new action "A" starts (assuming continuous time, with no discrete jumps from "t" to "t+1")?

We have pointed out on II 133 of the Supplement that biological measurements in mice find a delay in the range of 30 – 60ms before efferent motor signals reach sensory cortices. This interval defines implicitly the duration of a discrete time step in the context of our abstract model, and the length of a unit delay.

To the best of our knowledge it is commonly assumed in models for learning goal directed behavior and planning in the brain (see Mattar and Lengyel, Neuron 2022) that the brain “knows” when one action is finished, and a new action starts. We follow this common convention. There are reports that the basal ganglia are involved in chunking behavior, but this is obviously far beyond the topic of our ms.

(3) Neuromorphic hardware: The clarification that the model could be implemented in digital neuromorphic processors is very useful. However I believe that also the IBM chips that use memristor arrays operate in discrete/clocked time. If this is the case the authors should specify this, because otherwise using the term analog might lead the reader to believe that the implementation of the model would work in continuous time.

We have added on I 499 a remark which clarifies this point

(4) Energy efficiency: the authors should be very careful with claims on low power. For example, if you read carefully the Loihi paper mentioned, you will find claims such as "Neuromorphic computing *aims* to [...] at low power levels." in the abstract. But if you search for concrete numbers, you will find figures of >10 Watts for *dynamic* power (i.e., removing all the substantial power consumption figures of all the circuits that burn power even when the network is not running).

The claim of the reviewer that only dynamic power is measured in the cited literature is incorrect. For example, in our publication

Rao, A., Plank, P., Wild, A., & Maass, W. (2022). A long short-term memory for AI applications in spike-based neuromorphic hardware. *Nature Machine Intelligence*, 4(5), 467-479.

we measured static and dynamic power. The concrete numbers can be found in the supplement tables S2 and S4 (https://static-content.springer.com/esm/art%3A10.1038%2Fs42256-022-00480-w/MediaObjects/42256_2022_480_MOESM1_ESM.pdf).

We used the total power (static + dynamic) to calculate the energy and EDP ratios shown in the main text of the paper.

Even so, this *dynamic* figure is still much lower power than GPUs and cloud servers, but it is much higher power than a dedicated digital ANN accelerator for deep networks. The latest ANN accelerators can demonstrate high accuracy classification with less than 1W and sometimes even in the mW range. So even if *in principle* neuromorphic HW could be designed to be very low power, the SpiNNaker and Loihi platforms should not be listed as "low power" chips.

Well, the measurements of dynamic AND STATIC power in

Rao, A., Plank, P., Wild, A., & Maass, W. (2022). A long short-term memory for AI applications in spike-based neuromorphic hardware. *Nature Machine Intelligence*, 4(5), 467-479.

show that the Loihi chip which was used there for solving the sMNIST task was operating in the mW range. Hence we do not understand why one cannot refer to that chip as a "low power" chip.

To their defense, they were not designed to minimize power consumption. Indeed, a full SpiNNaker system with 1200 boards should consume about 90 kW.(4) 8-bit vs 4-bit precision: as for claims on energy efficiency the authors should be careful when citing papers and figures on equivalent bit-resolution of memristive devices. It is well known that memristive devices have high variability and cannot achieve 8-bit resolution during learning. The paper that the authors cite from the group of Joshua Yang is achieving a remarkable result, but only thanks to a "denoising" process applied only for inference when the true desired weight matrix is known. This is an

iterative process involving multiple reads and multiple writes that would not work in an on-line learning setup.

We are grateful for this correction. We are now referring instead on I 525 to Li, C., Hu, M., Li, Y., Jiang, H., Ge, N., Montgomery, E., ... & Xia, Q. (2018). Analogue signal and image processing with large memristor crossbars. *Nature electronics*, 1(1), 52-59.

It had been shown there that 8-bit weight resolution can be achieved in memristors without a denoising process.

Also other techniques (e.g. with the Hermes chip) will require extra operations and overhead to achieve 8-bit resolution figures. So the authors should be more cautious in making claims about 8-bit implementations in hardware that has memristive devices.

Well, the reference above does demonstrate 8-bit resolution of memristors. So this is to the best of our knowledge, and the knowledge of experts that we consulted, the state of the art. Obviously, further improvements of memristor technology can be expected in view of the tremendous research effort in this area.

The model proposed is already very strong on its own grounds. There is no need to make bold statements that might not be accurate if one were to actually try to implement it in hardware.

We are not aware of any remaining inaccurate statement in our paper after correcting the reference above.

(5) Transformers comparison: I read the authors arguments and agree with them on this point.

In summary, overall the paper has improved significantly. Despite the long review discussion, I believe there are very few minor edits to add to the already very strong and convincing manuscript, to address the issues raised above.

Reviewer #2 (Remarks to the Author):

The revised manuscript answers most of my initial concerns. A few remaining suggestions:

Line 165, "they do not depend on any specific policy or goal": But Eqn 4 explicitly depends on the goal o^* .

Thank you pointing this out. We corrected the line to:

“But in contrast to value estimates, they do not depend on a policy and are simultaneously available for every possible goal.”

Line 219, “affordance”: First appearance of the term in the text. Needs a definition in the present context.

Good point. We added an explanation on ll 217.

Line 228, “affordance values arise in the brain from a mixture nature”: Meaning? Sorry, we meant ...from a mixture of nature and nurture.

Figs 4 and 5: I had flagged the use of emojis in Figs 4 and 5 and the supplement as a distraction. In response the authors write (line 409), "we did not label different observations by numbers or letters because this might suggest a linear order of the observations, which does not exist". This is hard to understand because
a) Just about every paper about graphs indexes the nodes with integers, without implying any order.

But these papers are probably not addressing the problem of learning the graph structure through exploration, as we do here.

b) When labeling the actions A, B, C, D, the authors don't seem to have the same concern.

Agreed. We are not perfect.

c) The emojis have exactly the opposite effect: the reader is immediately led to inspect the pictures for any meaning, in particular whether two pictures are the same.

The latter is the intended function of this pictures when they are used in behavioral experiments in cognitive science. We aim to duplicate their functional role in our figures.

d) In the rebuttal the authors refer to Whittington 2020, a paper that makes copious use of emojis. My students just nominated that paper in a "5 worst figures" contest, so I don't consider it a good role model.

At the risk of ending up on the same list of the students of the reviewer, we have decided after a long discussion to stick to our way of labeling “observations” by small images. We do not see them as emojis, since other readers will hopefully not attach emotional meaning to them. Similar small images are actually commonly used by cognitive scientists in behavioral experiments if they just want the subject to note that certain screen contents are pairwise different. On the other hand, I never heard that a cognitive scientist used numbers instead. Since our paper also addresses readers from cognitive science, especially those who are interested in cognitive maps, we rather respect this tradition.

We want to point out that the earlier paper on cognitive maps Behrens, T. E., Muller, T. H., Whittington, J. C., Mark, S., Baram, A. B., Stachenfeld, K. L., & Kurth-Nelson, Z. (2018). What is a cognitive map? Organizing knowledge for flexible behavior. *Neuron*, 100(2), 490-509 should for fairness also be placed on the student’s list, in spite of the fact that it is a standard reference on cognitive maps with 755 citations: It also employs this labeling convention in Fig. 8D.

Also

Peer, M., Brunec, I. K., Newcombe, N. S., & Epstein, R. A. (2021). Structuring knowledge with cognitive maps and cognitive graphs. *Trends in cognitive sciences*, 25(1), 37-54 should be added because of their Fig. 1

Line 413: “Note that this encoding of actions is actionable”. Meaning? Please explain, or use different words.

Since we found on a 2nd look that this term is used by different authors with different meaning, we simply deleted it. It does not appear to offer additional insight.

Fig 4a: My suggestion “use different arrows for the 4 actions” meant the arrows should point right, up, left, down. Not different hash marks.

We beg to differ. Letting them point up, down, etc suggests innate knowledge about the impact of these actions in a 2D state. But the CML is not endowed with such innate knowledge. Our current way of labeling them by different line textures and letters appears to better convey this lack of information about their functional impact in 2D spaces.

Typos:

Line 425, “tested in on” Fixed

Line 950 Fixed

Supplement lines 26-27 Fixed